# Personalized Additive Modeling for Multi-level Federated Learning

Shutong Chen[1]  Guodong Long[1]  Tianyi Zhou[2]  Jie Ma[1]  Jing Jiang[1]  Chengqi Zhang[3]

## Abstract

Contemporary AI faces the challenge of balancing generality with user-specific personalization. In federated learning (FL), this challenge is amplified by highly heterogeneous client data with complex non-IID patterns beyond standard IID assumptions. Many existing FL methods are designed for relatively restricted heterogeneity settings (e.g., a fixed number of clusters or a fixed form of personalization), limiting their robustness under complex structures. In this work, we study FL from a *multi-level non-IID* perspective, where client similarity is captured by multiple granularities of shared knowledge: global, subgroup, and client-specific components. This view captures coarse-to-fine relationships while requiring less prior knowledge of task boundaries. Building on this insight, we propose *Federated Multi-level Additive Modeling* (FeMAM), which learns multiple levels of shareable models and constructs personalized predictors via additive composition across levels. To move beyond a fixed structure, FeMAM allows models to grow and be pruned dynamically during training, adapting to diverse federated scenarios. Despite employing multiple models, FeMAM remains cost-friendly by unlocking only a small subset (one level) of models for training at a time. Extensive experiments show that FeMAM effectively approximates diverse complex non-IID structures and consistently outperforms representative clustered and personalized FL baselines.

Code: https://github.com/shutong043/FeMAM

## 1. Introduction

The rapid expansion of contemporary AI systems has sharpened the tension between *generalization* and *personaliza-*

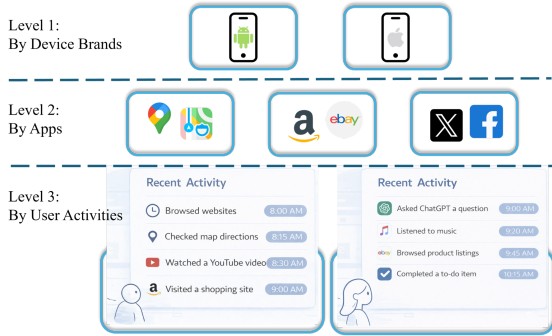

*Figure 1.* A multi-level structure for fine-grained service provision to smartphone users.

*tion*. Federated learning (FL) is increasingly adopted to enable privacy-preserving personalization, yet it also magnifies client heterogeneity. Importantly, as modern AI moves toward *large-scale, multi-capability* models (e.g., foundation models with client-side adaptation), non-IID effects become more structured and complex than those assumed by many standard FL formulations. This raises new requirements for FL algorithms: they should support *multi-granular sharing* and *fine-grained personalization* under complex non-IID conditions.

A key difficulty is that heterogeneity structure is often *unknown* and *implicit*, and can be of multiple levels. As illustrated in Fig. 1, consider a smartphone app service provider aiming to offer fine-grained services to users. At a coarse level, users can be grouped by operating systems such as Android and iOS. At a finer level, users may share subsets of commonly used applications, and at an even finer level, users can be differentiated by their activity patterns. Additional levels may naturally emerge when richer contextual or behavioral information is available. This motivates viewing FL through a *multi-level non-IID* lens, where client similarity can be approximated by multiple granularities of shared knowledge, ranging from global commonality to subgroup-level patterns and finally client-specific traits.

Many existing clustered FL (Ghosh et al., 2020; Briggs et al., 2020; Ma et al., 2022) and personalized FL (PFL) methods (Kairouz et al., 2021; Li et al., 2020a; Charles et al., 2024; Li et al., 2025b; 2026a) are designed around *restricted* or *pre-specified* heterogeneity assumptions—e.g.,

[1]University of Technology Sydney, Australia [2]Mohamed bin Zayed University of Artificial Intelligence, UAE [3]The Hong Kong Polytechnic University, China. Correspondence to: Guodong Long <guodong.long@uts.edu.au>.

*Proceedings of the 43rd International Conference on Machine Learning*, Seoul, South Korea. PMLR 306, 2026. Copyright 2026 by the author(s).

a fixed number of clusters, a fixed global-plus-local personalization form, or a fixed pipeline. As a result, they can struggle when client relationships exhibit richer, structured non-IID patterns induced by modern personalization needs. Many of them capture at most two sharing granularities (e.g., global → cluster, or global → client), and typically require prior knowledge of task boundaries or cluster structure. In realistic settings where the appropriate granularity (and even the number of granularities) is unknown a priori, an FL system should avoid committing to a fixed hierarchy and instead *adapt* its structure to the data. Several works have explored incorporating multi-level or hierarchical structure into FL, such as extracting multi-level branches with distillation (Kim et al., 2022), multi-level prototype regularization (Guo et al., 2024), or combining global modeling, clustering, and fine-tuning in a fixed pipeline (Zhang et al., 2024; Li et al., 2025a). While these approaches demonstrate the potential of multi-level ideas, they often rely on pre-defined hierarchies or fixed clustering strategies, limiting flexibility under diverse and evolving non-IID scenarios.

To address these limitations, we propose *Federated Multi-level Additive Modeling* (FeMAM), a flexible framework designed for complex, multi-granular non-IID heterogeneity. FeMAM organizes knowledge into a hierarchy of models: higher-level models capture coarse-grained patterns shared across broader groups, while lower-level models progressively encode finer-grained, subgroup- or client-specific information. Each client constructs its personalized predictor by *additively composing* one selected model from each level, enabling coarse-to-fine personalization without requiring explicit task boundaries. Crucially, FeMAM does not assume the hierarchy depth or structure is known in advance: it *grows* the hierarchy progressively in a boosting-like manner and *prunes* unnecessary models based on validation improvement. Despite employing multiple models, FeMAM remains *cost-friendly* by activating and communicating only one level of models for training at a time while freezing previously learned levels, making it scalable for realistic federated deployments.

This paper makes the following contributions:

- We formulate federated personalization under *multi-level non-IID* heterogeneity, motivated by the increasing structural complexity induced by contemporary AI personalization.
- We introduce a *multi-level additive* personalization mechanism that constructs client predictors via residual-style composition across levels.
- We propose a *grow-and-prune* training pipeline that expands the hierarchy progressively while pruning unnecessary levels per client, achieving adaptivity without committing to a fixed structure.

- We provide a convergence analysis for the proposed multi-level framework.
- Extensive experiments demonstrate that FeMAM consistently outperforms strong clustered and personalized FL baselines under challenging non-IID settings.

**Conflict of Interest Disclosure** The authors declare no financial conflicts of interest.

## 2. Related Works

**Single-model FL.** Vanilla federated learning typically refers to FedAvg (McMahan et al., 2017), where a single global model is learned by averaging local updates. Prior works improve FedAvg's convergence speed (Li et al., 2020b; Karimireddy et al., 2020) and analyze its behavior under non-IID data (Li et al., 2019). FedAvg is also widely used as a foundation for methods that build additional personalized components on top of a global model. In FeMAM, we leverage a global model as the coarsest (Level 1) shared component.

**Clustered FL.** Clustered FL (Guo et al., 2025; Fenoglio et al., 2025) assumes clients share knowledge within several clusters and aggregates within each cluster separately. CFL (Sattler et al., 2020) recursively splits clusters when gradient divergence is large. Other approaches adopt EM-style optimization (Xie et al., 2021) or hierarchical clustering (Briggs et al., 2020), with convergence analysis in (Ma et al., 2022). IFCA (Ghosh et al., 2020) assigns clients by selecting the cluster model that achieves the best local performance. More recent work (Bao et al., 2023) employs two-phase optimization to decouple clustering and model training. These methods typically model one intermediate sharing granularity (global → clusters), while FeMAM generalizes to multiple hierarchical levels and allows clients to compose information across levels.

**Personalized FL.** Personalized FL (Wu et al., 2023; Yu et al., 2024; Tran et al., 2025; Chen et al., 2025; Li et al., 2026b; Chen et al., 2026) aims to learn client-optimal models simultaneously. pFedMe (T Dinh et al., 2020) uses Moreau envelopes to decouple personalized optimization from the global model. Ditto (Li et al., 2021a) optimizes global and local models in a bi-level manner with regularization. Other methods personalize parts of the model, such as prototypes (Tan et al., 2021), head layers (Collins et al., 2021), or batch normalization layers (Li et al., 2021b). (Yi et al., 2024) uses personalized gates to combine global and local representations. Many PFL methods can be interpreted as two-level sharing (global + client), while FeMAM extends this view to a general multi-level hierarchy.

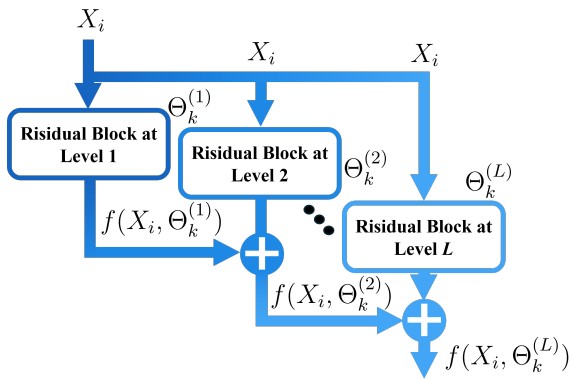

*Figure 2.* A client-specific personalized FL model is implemented as an additive of residual blocks across levels.

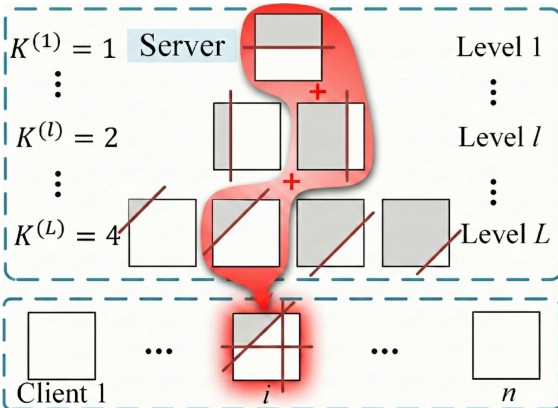

*Figure 3.* The architecture of multi-level federated learning with personalized additive modeling. The $i$-th client's model is a personalized additive modeling of multiple shared global models from different levels.

**Multi-level structured FL.** A few works relate to multi-level ideas in FL (M Ghari and Shen, 2024; Li et al., 2025a). Kim et al. (2022) treat networks of different depths as different levels and align outputs across levels. Guo et al. (2024) regularize prototypes across multiple levels. Zhang et al. (2024) combines separate modules to utilize resources at different levels. Compared with these designs (often tied to specific architectures or fixed pipelines), FeMAM provides a general additive modeling view that can flexibly incorporate different level-wise FL strategies.

**Additive modeling in FL.** Additive modeling was introduced in (Agarwal et al., 2020), where the local model is treated as a residual to the global model. (Ma et al., 2023) applies additive modeling to clustered FL by integrating a global model and cluster models. (Li et al., 2023) adds local models to global models to complement missing features. These methods typically employ a fixed two-level additive structure, while FeMAM generalizes additive residual composition to *multiple* levels to approximate richer structured non-IID patterns.

## 3. Methodology

### 3.1. Learning in Federated Settings

In a federated learning system, there are $m$ clients collaboratively learning model parameters. Client $i$ has a private dataset $D_i$ with $n_i$ instances, and $n = \sum_{i=1}^{m} n_i$ denotes the total number of instances. Let $\ell(\cdot)$ denote the loss measuring the prediction error between ground truth $Y_i$ and the model prediction $f(X_i; \Theta)$. FedAvg (McMahan et al., 2017) optimizes the global objective:

$$\min_{\Theta} \sum_{i=1}^{m} \frac{n_i}{n} \ell(Y_i, f(X_i; \Theta)). \qquad (1)$$

Under severe non-IID heterogeneity, a single global model

may fail to represent diverse client behaviors, motivating structured sharing and personalization.

### 3.2. Client-side Additive Residual Modeling

We model multi-level heterogeneity by introducing a hierarchy of shared models across $L$ levels. At level $l \in \{1, \ldots, L\}$, the server maintains $K_l$ models $\{\Theta_k^{(l)}\}_{k=1}^{K_l}$, representing knowledge at granularity $l$ (coarse-to-fine as $l$ increases). As illustrated in Fig. 3, each client constructs its personalized predictor by selecting (at most) one model per level. Then, for each client, inspired by residual learning (He et al., 2016), the predictions of these selected models are additively composed to form the client's final prediction, as shown in Fig. 2.

Formally, let $r_{i,k}^{(l)} \in \{0, 1\}$ denote the assignment indicator: $r_{i,k}^{(l)} = 1$ if client $i$ selects model $k$ at level $l$. The personalized predictor of client $i$ is:

$$f_i(X_i) = \sum_{l=1}^{L} \sum_{k=1}^{K_l} r_{i,k}^{(l)} f(X_i; \Theta_k^{(l)}). \qquad (2)$$

Intuitively, the additive stacking across levels in Fig. 2 yields a coarse-to-fine refinement path, where higher-level components provide shared structure and lower-level components act as residual refinements.

### 3.3. Server-side Level-wise Aggregation

Given client assignments at a level (as routed in Fig. 3), the server aggregates models in a level-wise manner, with no cross-level aggregation between different levels of models. Specifically, for level $l$ and model $k$, the aggregated model

is:

$$\Theta_k^{(l)} \leftarrow \frac{\sum_{i=1}^m n_i \, r_{i,k}^{(l)} \, \theta_i^{(l)}}{\sum_{i=1}^m n_i \, r_{i,k}^{(l)}}, \qquad (3)$$

where $\theta_i^{(l)}$ denotes the local copy trained by client $i$ for its selected model at level $l$.

## 3.4. Overall Objective and Assignment

Our overall objective is to learn the hierarchy of models and the client-specific assignment path:

$$\min_{\{r;\Theta\}} \sum_{i=1}^m \frac{n_i}{n} \ell\left(Y_i, \sum_{l=1}^L \sum_{k=1}^{K_l} r_{i,k}^{(l)} f(X_i; \Theta_k^{(l)})\right). \quad (4)$$

In practice, $r_{i,k}^{(l)}$ is determined by a level-wise assignment rule. A general form is:

$$r_{i,k}^{(l)} \leftarrow \mathbb{I}\left[k = \arg\min_{k'} \mathcal{L}\left(D_i; \Theta_{k'}^{(l)}\right)\right], \qquad (5)$$

where $\mathcal{L}$ is an assignment criterion (e.g., validation loss, distance to prototypes, or cluster likelihood). As shown later in our experiments (Table 2), FeMAM can instantiate different FL strategies at different levels (e.g., FedAvg at Level 1, K-Means/clustered FL at intermediate levels, and local fine-tuning at the finest level). The framework is not limited to specific choices and can flexibly integrate existing FL methods to match various structured non-IID scenarios. The instantiated objective and assignment rules for Table 2 are given in Appendix C.

## 3.5. Boosting-like Level-wise Optimization for Efficiency and Flexibility

Directly optimizing all $L$ levels jointly is costly and requires pre-specifying the hierarchy depth, which is often unknown in practice. Inspired by gradient boosting (Friedman, 2001), FeMAM learns levels sequentially: it gradually adds new levels and optimizes only the latest level while freezing all previously learned levels. This yields stable training and makes the method cost-adaptive. With this level-wise strategy, each round communicates and optimizes only the current level's model per client, while earlier levels serve as fixed residual components. Concretely, the client update at the current level $l$ is:

$$\theta_i^{(l)} \leftarrow \theta_i^{(l)} - \eta \nabla_{\theta_i^{(l)}} \ell\Big(Y_i, \underbrace{\sum_{l'=1}^{l-1} \sum_{k=1}^{K_{l'}} r_{i,k}^{(l')} f(X_i; \Theta_k^{(l')})}_{\text{frozen previous levels}}$$
$$+ \underbrace{\sum_{k=1}^{K_l} r_{i,k}^{(l)} f(X_i; \theta_i^{(l)})}_{\text{current level}}\Big). \qquad (6)$$

---

**Algorithm 1:** Federated Multi-Level Additive Modeling (FeMAM)

---

**Input** : Training & validation data, maximum level $L$.
**Output** : Models $\{\Theta_k^{(l)}\}$ and assignments $\{r_{i,k}^{(l)}\}$ for levels 1 to $L$.

1 **for** $l = 1$ **to** $L$ **do**
2    Initialize models $\{\Theta_k^{(l)}\}_{k=1}^{K_l}$ and assignments $\{r_{i,k}^{(l)}\}$;
3    **while** *not converge at level $l$* **do**
     /* Client updates (in parallel) */
4      **foreach** *client $i$* **do**
5        Receive assigned model at level $l$ (and keep levels $< l$ frozen);
6        Update local model $\theta_i^{(l)}$ using Eq. (6);
7        Upload $\theta_i^{(l)}$ to server;
     /* Server updates */
8      Update assignments $\{r_{i,k}^{(l)}\}$ via Eq. (5);
9      Aggregate level-$l$ models via Eq. (3);
10      Broadcast updated level-$l$ models to clients;
   /* Client-side structure pruning */
11    **foreach** *client $i$* **do**
12      Accept level $l$ only if adding its output reduces validation error (Eq. (7));

---

This boosting-like optimization provides:

- **Training stability:** higher-level models are learned on top of converged lower-level components. Freezing previous levels makes each new level learn a residual correction, which reduces interference and improves stability under non-IID.
- **Cost adaptivity:** training can stop early once performance saturates, or client capacity is limited.
- **Communication-friendly:** only one model (current level) is sent per client per round, keeping per-round communication the same as single-model FL while progressively enabling multi-level personalization.

### 3.6. Structure Pruning

Similar to the decision tree, as the hierarchy grows, overly long additive paths may overfit or become unnecessary for some clients. Therefore, FeMAM applies client-specific pruning: a client keeps a new level only if it improves validation performance. As shown in line 11 and 12 in Algorithm 1, after finishing training level $l$, client $i$ accepts it if:

$$\ell(Y_i, f_i^{(<l)}(X_i)) > \ell\Big(Y_i, f_i^{(<l)}(X_i) + \sum_{k=1}^{K_l} r_{i,k}^{(l)} f(X_i; \Theta_k^{(l)})\Big). \quad (7)$$

where $f_i^{(<l)}(X_i)$ denotes the composed predictor using the accepted levels $1, \dots, l-1$. This pruning ensures that additional parameters introduced by FeMAM are always

utility-improving for each client, yielding an on-demand personalization depth.

## 4. Discussion and Analysis

### 4.1. Discussion of $L$ and $K$

In practice, the maximum level $L$ does not need to be fixed by the server to match all clients. Instead, FeMAM treats personalization depth as client-dependent: due to the structure pruning technique, a client will stop accepting new levels once performance saturates. In our experiments (Table 2), we set a maximum level of 5, which is sufficient to accommodate the simulated non-IID settings.

Clustering in FeMAM can be viewed as a sampling strategy for inherent structures. More clustered levels correspond to sampling more possible structures (i.e., possible model assignments $r$), while structure pruning determines which sampled structures are effective. Therefore, as long as $K$ is set within a reasonable range (i.e., $1 < K < m$), a sufficient number of clustered levels can approach the same inherent structure. In contrast, prior clustered FL methods are fundamentally different from FeMAM, rely on a fixed number of clusters and thus require careful tuning to match specific non-IIDness. Following common clustering settings where $K$ typically ranges from 3 to 7, we adopt a simple configuration and fix $K = 5$ across all simulated scenarios and large-scale settings in our experiments.

### 4.2. Cost Analysis

FeMAM maintains multiple levels of models, which may appear resource-intensive at first glance. However, FeMAM is designed to be scalable and deployment-friendly through level-wise training and pruning.

**FeMAM adapts to heterogeneous client resources.** Because models are added sequentially and clients can stop accepting new levels at any time, FeMAM naturally supports both resource-constrained clients (accepting fewer levels) and resource-rich clients (using deeper personalization paths). As a result, FeMAM can maximize each client's performance under its own cost constraints.

**Training heads with a shared extractor.** As an additional practical implementation for efficiency, different levels can train only lightweight task-specific heads while sharing a common feature extractor; we *do not rely on this design* here, as FeMAM is intended as a general framework not restricted to neural network architectures.

**Training and inference cost.** From a system perspective, federated learning cost mainly arises from communication and client-side computation. During training, FeMAM communicates and optimizes only one current-level model per client per round, keeping per-round communication the same as single-model FL. At inference time, a client may store up to $L$ models, leading to at most an $L$-fold increase in computation compared to single-model methods. However, since models from different levels can be evaluated in parallel, the inference latency can be close to that of a single model. Moreover, structure pruning typically results in far fewer than $L$ accepted levels in practice (Table 4).

### 4.3. Theoretical Analysis

**Convergence intuition.** Let $\mathcal{R}_l$ be the objective in Eq. (4) restricted to the first $l$ levels, i.e., the loss of the additive predictor in Eq. (2) when only levels $1, \ldots, l$ are active. Our analysis shows two simple facts. First, for a fixed level $l$, optimizing Eq. (4) with respect to $\Theta^{(l)}$ (while freezing lower levels) yields a standard averaged-gradient convergence bound for $\mathcal{R}_l$. Second, when a new level is added, structure pruning together with a boosting-style residual alignment guarantees that $\mathcal{R}_l \leq \mathcal{R}_{l-1}$. As a result, in the same idea as gradient boosting—where models are added sequentially to reduce the remaining residual—the sequence of objectives corresponding to the progressively enriched models is monotone non-increasing:

$$\mathcal{R}_L \ \leq \ \mathcal{R}_0 \ - \ \sum_{l=1}^{L} \frac{\gamma_l^2}{2\beta} \left\| \nabla \mathcal{R}_{l-1} \right\|_2^2 \tag{8}$$

which reflects the boosting intuition: each newly added level is trained to align with the residual induced by earlier levels, and $\gamma_l \in (0, 1]$ measures the degree of this alignment at level $l$. For full convergence analysis please refer to Appendix D.

## 5. Experiments

### 5.1. Simulated Scenarios

We evaluate FeMAM under a range of controlled federated learning scenarios with diverse and complex non-IID settings, in order to assess its adaptability and performance.

#### 5.1.1. EXPERIMENTAL SETUP

**Datasets and system settings.** We evaluate all methods on two benchmark image classification datasets: Tiny ImageNet (Le et al., 2015) and CIFAR-100 (Krizhevsky et al., 2009). We simulate an FL system with $m = 50$ clients. Each communication round performs 2 local epochs. We use ResNet-9 (He et al., 2016) as the base model.

**Non-IID partitions.** We consider three non-IID settings designed to cover both *single-level* and *multi-level* heterogeneity:

**(I) Cluster-wise non-IID.** This is a common setting for clustered FL. We split the label space into several clusters to

*Table 1.* Test results (mean±std over 4 runs) on Tiny ImageNet and CIFAR-100 under three non-IID partitions.

| Dataset | Non-IID Type | Metric | Local | FedAvg | FedAvg+ | Ditto | FeSEM | FedEM | pFedMoE | FedCAM | PM-MoE | FeMAM |
|---|---|---|---|---|---|---|---|---|---|---|---|---|
| Tiny ImageNet | Cluster-wise (50,5) | Accuracy | 15.57±0.03 | 24.87±0.80 | 36.74±0.09 | 37.12±0.70 | 41.20±0.25 | 42.61±0.23 | 39.82±0.57 | 46.32±0.55 | 27.37±0.48 | **46.80±0.35** |
| | | Macro-F1 | 10.42±0.17 | 12.63±0.49 | 32.97±0.23 | 33.74±0.80 | 37.82±0.21 | 39.92±0.30 | 36.81±0.62 | 42.76±0.84 | 23.69±0.41 | **43.29±0.34** |
| | Client-wise $\alpha = 0.1$ | Accuracy | 28.93±0.21 | 24.33±0.34 | 45.85±0.13 | 47.08±0.16 | 22.34±0.79 | 28.75±0.53 | 49.88±0.20 | 33.64±0.24 | 42.29±0.32 | **60.46±0.35** |
| | | Macro-F1 | 8.00±0.15 | 8.73±0.24 | 23.84±0.03 | 24.57±0.23 | 7.79±0.19 | 10.34±0.15 | 29.14±0.13 | 13.55±0.28 | 20.48±0.29 | **33.29±0.15** |
| | Multi-level [9,15,22] | Accuracy | 39.48±0.36 | 17.91±0.09 | 56.80±0.21 | 58.83±0.18 | 34.14±0.69 | 37.34±0.28 | 59.11±0.12 | 34.78±0.22 | 55.66±0.30 | **65.30±0.17** |
| | | Macro-F1 | 28.99±0.43 | 4.49±0.07 | 52.33±0.56 | 51.82±0.53 | 19.10±0.67 | 20.85±0.36 | 54.00±0.13 | 17.07±0.51 | 49.59±0.34 | **60.46±0.35** |
| CIFAR-100 | Cluster-wise (30,5) | Accuracy | 22.81±0.11 | 29.28±0.18 | 46.67±0.74 | 45.16±0.18 | 45.10±0.51 | 48.31±0.89 | 49.03±0.35 | 53.29±0.69 | 52.61±0.41 | **53.72±0.29** |
| | | Macro-F1 | 13.80±0.33 | 12.73±0.15 | 39.77±0.91 | 38.05±0.76 | 38.73±0.60 | 44.05±0.31 | 41.83±0.27 | 47.89±0.17 | 46.71±0.36 | **48.33±0.40** |
| | Client-wise $\alpha = 0.1$ | Accuracy | 38.57±0.15 | 30.46±0.15 | 53.57±0.45 | 56.06±0.48 | 25.88±0.12 | 32.25±0.30 | 57.76±0.30 | 37.47±0.18 | 60.10±0.44 | **61.96±0.48** |
| | | Macro-F1 | 11.90±0.11 | 11.53±0.08 | 29.49±0.41 | 30.71±0.34 | 9.42±0.18 | 13.88±0.17 | 33.26±0.56 | 14.52±0.10 | 34.02±0.39 | **36.41±0.70** |
| | Multi-level [3,6,11] | Accuracy | 28.67±0.33 | 25.08±0.30 | 49.73±0.32 | 51.73±0.39 | 41.48±0.59 | 46.89±0.97 | 56.70±0.23 | 51.53±0.24 | 59.44±0.37 | **61.86±0.32** |
| | | Macro-F1 | 11.97±0.17 | 15.06±0.15 | 40.13±0.47 | 42.53±0.57 | 32.75±0.77 | 36.12±0.14 | 48.95±0.44 | 43.04±0.46 | 52.78±0.41 | **55.26±0.77** |

*(a)* Cluster-wise, $(30, 5)$ (Non-IID)  *(b)* Client-wise, $\alpha = 0.1$ (Non-IID)  *(c)* Multi-level, $[3, 6, 11]$ (Non-IID)  *(d)* IID

*Figure 4.* Convergence on CIFAR-100 under three non-IID partitions and the IID partition. FeMAM progressively adds new levels when the current additive model saturates, resulting in staircase-shaped improvements. Each blue vertical line marks the round where a new level is introduced.

*Table 2.* FeMAM settings in experiments. The number of level $L = 5$. The number of shared models $K$ for each level is $1, 5, 5, 5, 50$.

| Level ID | 1 | 2 | 3 | 4 | 5 |
|---|---|---|---|---|---|
| Number of Models (K) | 1 | 5 | 5 | 5 | 50 |
| Type of level | Global | Cluster | Cluster | Cluster | Personal |
| Level-wise Method | FedAvg | K-means | K-means | K-means | Local train |

form distinct distribution groups. As a tag, $(c, K)$ indicates $K$ clusters and each cluster contains $c$ classes (e.g., $(30, 5)$ means 5 clusters and 30 classes per cluster). This setting mainly represents non-IID with a *single cluster level*.

**(II) Client-wise non-IID.** We adopt the Dirichlet partition proposed by Hsu et al. (2019), which is widely used in personalized FL to model client-level heterogeneity. Specifically, data proportions across clients are sampled from a Dirichlet distribution with concentration parameter $\alpha$, and we set $\alpha = 0.1$ to simulate a highly heterogeneous setting. Although this partition is often treated as representing purely client-level (i.e., global + personal) non-IID, **our results suggest that it can exhibit implicit multi-level structure**.

**(III) Multi-level non-IID.** We explicitly construct a hierarchical non-IID structure with multiple levels. Each level contains a subset of the dataset, and data within that level is further divided into clusters. The number of clusters per level ranges from 1 to $m$ to simulate shared knowledge at different granularities. As a tag, $[3, 6, 11]$ means the dataset is split into three levels with 3, 6, and 11 clusters respec-

tively. Please refer to Appendix Table A.1 for details.

**Baselines.** For all baselines, we use a validation set to monitor performance and select the checkpoint with the minimum validation error over 700 communication rounds. We include representative methods from: (1) **Global FL:** FedAvg (McMahan et al., 2017); (2) **Clustered FL:** FeSEM (Xie et al., 2021), FedCAM (Ma et al., 2023); (3) **Multi-model FL:** FedEM (Marfoq et al., 2021) (with 5 shared models), and PM-MoE (Feng et al., 2025), a two-step method that retrains a personalized MoE router over classification heads after general federated learning. For PM-MoE, we use FedPer (Arivazhagan et al., 2019) as its first step; (4) **Personalized FL:** pFedMoE (Yi et al., 2024), Ditto (Li et al., 2021a), FedAvg+, and Local. FedAvg+ denotes client-side fine-tuning from the FedAvg global model, and Local trains each client independently using only local data. Detailed settings are provided in Appendix A.

**FeMAM instantiation.** We use the FeMAM hierarchy shown in Table 2 with maximum level $L = 5$. Level 1 is a FedAvg global model ($K_1 = 1$). Levels 2–4 use clustering-based learning with $K_l = 5$ (implemented via FeSEM with K-means assignments) to capture group-level knowledge. Level 5 performs local training with $K_5 = 50$ (one client-specific model per client) to capture fully personalized information. We run 400 communication rounds for Level 1; subsequent levels typically converge quickly (within around 50 rounds per level in our setup).

*Table 3.* Ablation on clustering levels $K$ (CIFAR-100, Client-wise non-IID).

| Method | FeMAM | | | | | | | pFedMoE |
|---|---|---|---|---|---|---|---|---|
| $K$ setting | (2,2,2) | (5,5,5) | (8,8,8) | (10,10,10) | (25,25,25) | (5,10,15) | (15,10,5) | |
| Accuracy (%) | 62.24 | 61.96 | 62.34 | 61.84 | 62.85 | 61.81 | 61.87 | 57.76 |

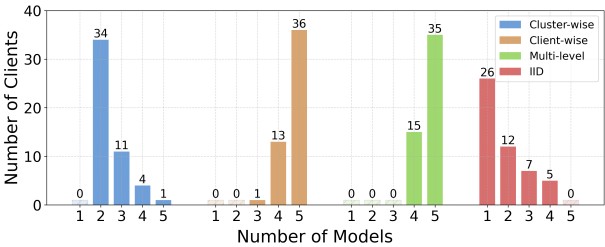

*Figure 5.* Model number distribution across clients. The total number of clients is 50. As an example, on Cluster-wise non-IID setting, 34 clients are assigned 2 models.

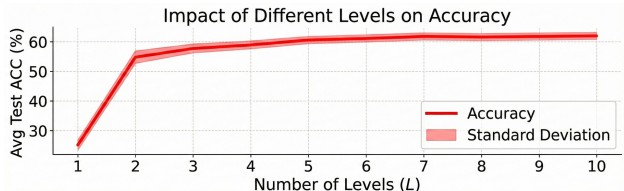

*Figure 6.* Impacts of different levels on accuracy. Accuracy consistently increases as $L$ increases, and the rate of improvement diminishes after $L$ exceeds 5.

### 5.1.2. NUMERICAL RESULTS

Table 1 compares FeMAM with baselines in terms of test accuracy and Macro-F1 (mean±std over 4 runs). We summarize two key observations. **(1) Most baselines are sensitive to the targeted non-IID structure.** Clustered FL methods (FeSEM and FedCAM) perform strongly on Cluster-wise non-IID but degrade substantially on Client-wise non-IID, where client distributions are highly individualized. Conversely, personalized FL methods are more competitive on Client-wise non-IID but may not consistently dominate in Cluster-wise settings.

**(2) FeMAM consistently achieves the best performance across all non-IID types.** FeMAM outperforms baselines on both single-level non-IID (Cluster-wise) and complex multi-level settings (Client-wise and Multi-level). This suggests that multi-level additive composition provides a robust mechanism for structured knowledge sharing across diverse federated scenarios.

### 5.1.3. CONVERGENCE ANALYSIS

**Adaptive behavior across data distributions.** Fig. 4 reports convergence curves under four data partitions. Fe-MAM consistently achieves strong performance with reasonable convergence speed. We use the same maximum hierarchy ($L = 5$; Table 2) across all settings, but the learned behavior differs due to structure pruning.

Specifically, under Client-wise and Multi-level non-IID, Fe-MAM exhibits up to five distinct plateaus, and each plateau corresponds to a performance improvement after adding a new level. In contrast, under Cluster-wise non-IID, FeMAM shows only two plateaus, indicating that later levels contribute marginally and are frequently rejected by structure

pruning. Under IID, FeMAM behaves similarly to FedAvg, and most clients retain only the first-level global model.

### 5.1.4. ANALYSIS OF MULTI-LEVEL STRUCTURE

**FeMAM discovers implicit structure via client-specific depth.** To understand how FeMAM adapts to the underlying heterogeneity, we visualize the distribution of accepted model counts across clients in Fig. 5.

By structure pruning, FeMAM assigns each client a different number of accepted models, reflecting the complexity of its local distribution. Under IID, more than half of the clients (26/50) keep only one model, indicating that a global model is sufficient. Under Client-wise and Multi-level non-IID, heterogeneity is stronger and most clients accept close to the maximum number of levels (often 5 models). Under Cluster-wise non-IID, most clients accept 2 models, which aligns with a single dominant cluster level. We further visualize representative learned structures in Appendix Fig. A.1.

### 5.1.5. ABLATION STUDY

**Choice of $L$.** Fig. 6 studies the impact of $L$ on Tiny ImageNet under Client-wise non-IID. We fix Level 1 as FedAvg and the final level as Local training, and vary the number of intermediate cluster levels. For example, $L = 7$ corresponds to 5 intermediate cluster levels. Accuracy improves consistently as $L$ increases, while the gains diminish beyond $L = 5$. Therefore, we set $L = 5$ as a practical trade-off between performance and model complexity.

**Choice of $K$.** Table 3 study the sensitivity of FeMAM to the number of clusters used in the intermediate clustered levels. We fix Level 1 as FedAvg and the final level as Local training, and vary the cluster numbers of the intermediate levels. Overall, FeMAM is not sensitive to the specific choice of $K$: across a wide range of cluster counts, perfor-

*Table 4.* Average number of stored models across clients at inference time.

| Methods | FedAvg | pFedMoE | FedCAM | FedEM | FeMAM |
|---|---|---|---|---|---|
| Cluster-wise | 1 | 2 | 2 | 5 | 2.4 |
| Client-wise | 1 | 2 | 2 | 5 | 4.7 |
| Multi-level | 1 | 2 | 2 | 5 | 4.7 |
| IID | 1 | 2 | 2 | 5 | 1.8 |

mance remains consistently strong. This behavior aligns with FeMAM's design: clustered levels mainly provide a set of candidate models, while client-side structure pruning selects only those components that reduce validation error. In this sense, clustering in FeMAM can be viewed as a sampling strategy for inherent multi-level structures: as long as $K$ is within a reasonable range ($1 < K < m$) and multiple clustered levels are available, FeMAM typically has sufficient choices to approximate the inherent structure. **This contrasts with prior clustered FL methods that rely on a fixed cluster number to directly fit the non-IID structure, and thus can be much more sensitive to $K$.**

**Efficient training time.** Fig. 7a compares wall-clock runtime. FeMAM is only moderately slower than FedAvg and is faster than Ditto, since it activates only the latest level each round (one model per client), while Ditto trains two models per client per round.

**Efficient Communication.** Fig. 7b plots test accuracy against communicated parameter volume. FeMAM achieves strong accuracy with acceptable communication cost because only one current-level model is uploaded/downloaded per client per round.

**Adaptive inference cost.** The inference cost of FeMAM is proportional to the number of stored models per client. We measure it by the average number of models retained for inference. As shown in Table 4, FeMAM typically retains fewer than 5 models per client through pruning, resulting in lower storage cost than FedEM.

## 5.2. Large-Scale, Contemporary Scenarios

The generalization ability of foundation models enables richer client-specific tasks, thereby inducing more complex non-IID structures that go beyond traditional federated learning settings (Long, 2024; Yang et al., 2024; 2025b). We simulate such a setting and demonstrate that many baselines are not designed with such complex, task-driven heterogeneity in mind, while FeMAM's can effectively adapt to it. **Model and PEFT configuration.** Qwen3-4B (Yang et al., 2025a) is utilized as the base model with its thinking mode disabled. We adopt parameter-efficient fine-tuning (PEFT) via LoRA (Hu et al., 2021): clients share a common foundation model backbone, and federated learning is performed on LoRA parameters.

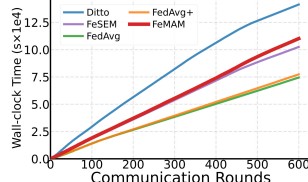 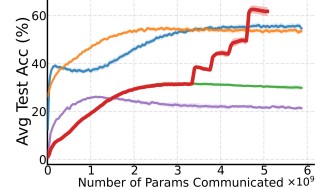

*(a)* Running time variation with increasing communication rounds. The running time of FeMAM is less than Ditto and greater than FedAvg.

*(b)* Performance variation with increasing communicate-parameter volume. FeMAM does not require a large number of communication parameters for high accuracy.

*Figure 7.* Comparison of FeMAM with other methods in terms of (a) running time and (b) communication efficiency.

**Clients and non-IID construction.** We construct a task-partitioned federated learning setting using tasks from LoraHub (Huang et al., 2023), derived from FLAN-T5 (Chung et al., 2024). We select 71 tasks with short sequence lengths and simulate 71 clients, where each client is assigned one task, inducing task-driven non-IID heterogeneity. Please refer to Appendix B for more details.

**Baselines.** We compare FeMAM with Local, FedAvg, FedAvg+, Ditto, FeSEM, FedEM, pFedMoE, and FedCAM, following the same configurations as in Section 5.1.1. PM-MoE is excluded as it is designed for classification-oriented settings and does not align with sequence-to-sequence generation tasks.

**Training and evaluation.** All methods are trained for 100 communication rounds with one local epoch per round. We select the checkpoint with the lowest validation loss and report ROUGE-1 as the evaluation metric.

### 5.2.1. RESULTS

**Numerica:** As shown in Table 5, FeMAM achieves both the lowest average loss and the highest ROUGE-1 score (mean over 4 runs). Cluster-based FL methods (e.g., FeSEM and FedCAM) generally achieve higher ROUGE-1 scores, while personalized FL methods such as FedAvg and Ditto tend to yield lower evaluation loss. This suggests that information sharing at different granularities is important for federated learning with foundation models. By integrating information sharing across multiple levels, FeMAM benefits from both aspects and achieves superior performance on both metrics.

**Structure: Beyond a simple global–local design.** We visualize the distribution of accepted model counts across clients in Fig. 8. Most clients utilize 3 or 4 models to solve their assigned tasks, and no client relies on only a single model. This observation reflects the inherent complexity of federated learning with foundation models, where multiple levels

*Table 5.* Results with foundation models on FLAN-T5 tasks.

|  | Local | FedAvg | FedAvg+ | Ditto | FeSEM | FedEM | pFedMoE | FedCAM | FeMAM |
|---|---|---|---|---|---|---|---|---|---|
| Loss | 1.160 | 1.135 | 1.095 | 1.087 | 1.114 | 1.110 | 1.140 | 1.117 | **1.040** |
| ROUGE-1 (%) | 53.90 | 55.25 | 55.19 | 56.11 | 58.65 | 56.54 | 54.73 | 58.00 | **59.25** |

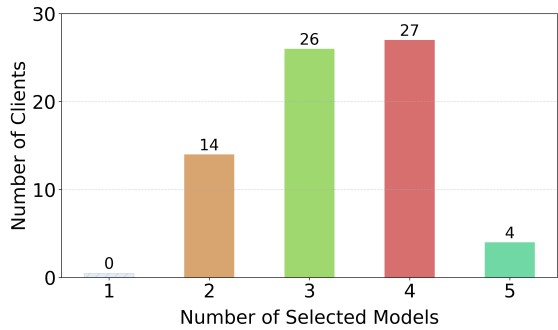

*Figure 8.* Model number distribution across clients in foundation model-based federated learning. Most clients utilize more than two models, indicating a naturally complex structure beyond a simple global–local design.

of shared and personalized components are often required.

## 6. Limitations and Future Works

**Limited exploration of level-wise FL methods.** FeMAM is a general multi-level framework that flexibly composes different federated learning paradigms for multi-granular sharing and personalization. Due to limited experimental budget, we mainly evaluate a representative instantiation (FedAvg at Level 1, K-means based clustered FL at intermediate levels, and local fine-tuning at the final level). We do not systematically study alternative level-wise components (e.g., IFCA, FedEM, Ditto, FedPer) or their cross-level interactions. A comprehensive study of which methods are most complementary at which levels—and whether such choices can be automated—is a promising direction that could further strengthen FeMAM's practicality and performance.

**Scale and task diversity in large-scale evaluation.** Our foundation-model experiments demonstrate FeMAM in a contemporary, task-driven heterogeneous setting, but our evaluation is still relatively small in scale compared to realistic deployments (relatively short and lightweight Flan-T5 tasks). In realistic deployments, heterogeneity can be more structured and multi-granular, as in the smartphone service scenario in Fig. 1 (OS-level commonality, subgroup similarity, and fine-grained personalization). Such nested sharing patterns are hard to pre-specify with fixed-depth designs. We expect FeMAM's additive, boosting-style hierarchy with grow-and-prune to better approximate coarse-to-fine structure, and thus widen the gap over level-limited methods

(e.g., global–local or single-cluster-level) in complex scenarios. An important future direction is to evaluate FeMAM at substantially larger scale with harder, more diverse tasks.

## 7. Conclusion

Real-world federated systems increasingly involve heterogeneous clients with complex and structured non-IID relationships. This paper proposes FeMAM, a multi-level federated learning framework that learns a hierarchy of shareable models and constructs client predictors via additive residual composition across levels. By combining level-wise training with client-side structure pruning, FeMAM provides a flexible mechanism to approximate diverse complex structures while remaining cost-adaptive: clients can stop at different depths depending on performance saturation or resource constraints. Extensive experiments across IID, single-level non-IID, and multi-level non-IID settings demonstrate the effectiveness and robustness of FeMAM.

## Impact Statement

This paper presents work whose goal is to advance the field of machine learning. There are many potential societal consequences of our work, none of which we feel must be specifically highlighted here.

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

# Appendix

# A. Additional Experimental Details and Analyses in Simulated Scenarios

### A.1. Hyper-parameter Settings

**FL system settings.** We simulate an FL system with $m = 50$ clients. Each communication round performs 2 local epochs with learning rate 0.01. We adopt ResNet-9 as the base model. We run all methods for up to 700 communication rounds in the main experiments.

**Train/validation/test splits.** We use a validation set to select checkpoints with the minimum validation error. The validation set is split from the original training set. We set the validation set size to be half of the test set size. Therefore, for CIFAR-100, the train/validation/test sizes are 45000/5000/10000, respectively. For Tiny ImageNet, the train/validation/test sizes are 95000/5000/10000, respectively.

### A.2. Data Partition Details

**Cluster-wise partition.** For Cluster-wise non-IID, each cluster is assigned a fixed set of classes. For example, $(30, 5)$ means five clusters and each cluster contains 30 classes. Data within each cluster is then evenly divided and assigned to clients belonging to that cluster.

**Multi-level partition.** For Multi-level non-IID, we construct a hierarchical partition inspired by Fig. 1. As shown in Table A.1, clients are grouped multiple times at different granularities. For example, on CIFAR-100, the three levels use 3, 6, and 11 clusters, respectively, yielding $3 + 6 + 11 = 20$ partitions in total. We split the original dataset into 20 non-overlapping label-based parts, and each part is evenly divided and assigned to the corresponding clients.

**Client-wise partition.** For Client-wise non-IID, we follow the Dirichlet partition commonly used in personalized FL. The concentration parameter controls the degree of heterogeneity, and in our experiments we set $\alpha = 0.1$.

| data-set | level id | class num | cluster num | data distribution structure |
|---|---|---|---|---|
| CIFAR-100 | 1 | 14 | 3 |  |
| | 2 | 6 | 6 | |
| | 3 | 2 | 11 | |
| Tiny ImageNet | 1 | 9 | 9 |  |
| | 2 | 5 | 15 | |
| | 3 | 2 | 22 | |

*Table A.1.* Data distribution structure for the multi-level partition. Each cell represents one client, and cluster boundaries are separated by black dashed lines.

### A.3. Method-specific Details

**Local.** Each client trains using only its local training and validation data.

**FedAvg and FedAvg+.** FedAvg trains a single global model via federated averaging. FedAvg+ denotes client-side fine-tuning from the FedAvg global model.

**Ditto.** We set the regularization coefficient to $0.1$.

**FeSEM.** The number of clusters is set to $5$.

**FedEM.** The number of shared models is set to $5$.

**pFedMoE.** Following Yi et al. (2024), we share a global backbone across clients while personalizing the classification head. Each client also maintains a local model, and a personalized gating network combines global and local representations

before the personalized head.

**FedCAM.** Following Ma et al. (2023), we adopt an additive clustered structure. We warm up with FedAvg for 5 rounds and initialize cluster-level models from the global model.

**FeMAM convergence and level-wise schedule.** To monitor convergence at each level, we track the variance of overall accuracy over the most recent 50 communication rounds. If the variance is below 1, we consider the current level converged. Empirically, with FedAvg as Level 1, later levels converge substantially faster. For consistent settings across non-IID partitions, we set the number of communication rounds to 400 for Level 1 and 50 for each subsequent level.

## A.4. More Analysis of Multi-level Structure

To further illustrate how FeMAM adapts to the underlying heterogeneity, we visualize the learned multi-level structures. We compare the learned structures with the ground-truth (pre-defined) relation structures, and include convergence curves for interpretation. We use Tiny ImageNet for illustration; results on CIFAR-100 are similar.

**Pruning redundant levels under simpler heterogeneity.** In Cluster-wise and IID settings (Fig. A.1a and Fig. A.1d), the ground-truth structure contains little high-level complexity. Correspondingly, FeMAM learns sparse structures at higher levels by rejecting models that do not improve validation performance.

**Discovering cluster boundaries under hierarchical heterogeneity.** For settings with explicit clusters, we mark a boundary when adjacent clients belong to different clusters. Under simple Cluster-wise non-IID, FeMAM discovers the corresponding cluster boundaries at Level 2. Under Multi-level non-IID, FeMAM reveals most boundaries across multiple levels, reflecting the coarse-to-fine organization. Note that in our default configuration, FeMAM uses $K_l = 5$ for Levels 2–4; therefore, it is not expected to perfectly match the ground-truth cluster counts (e.g., 9/15/22), but it still captures the major boundaries.

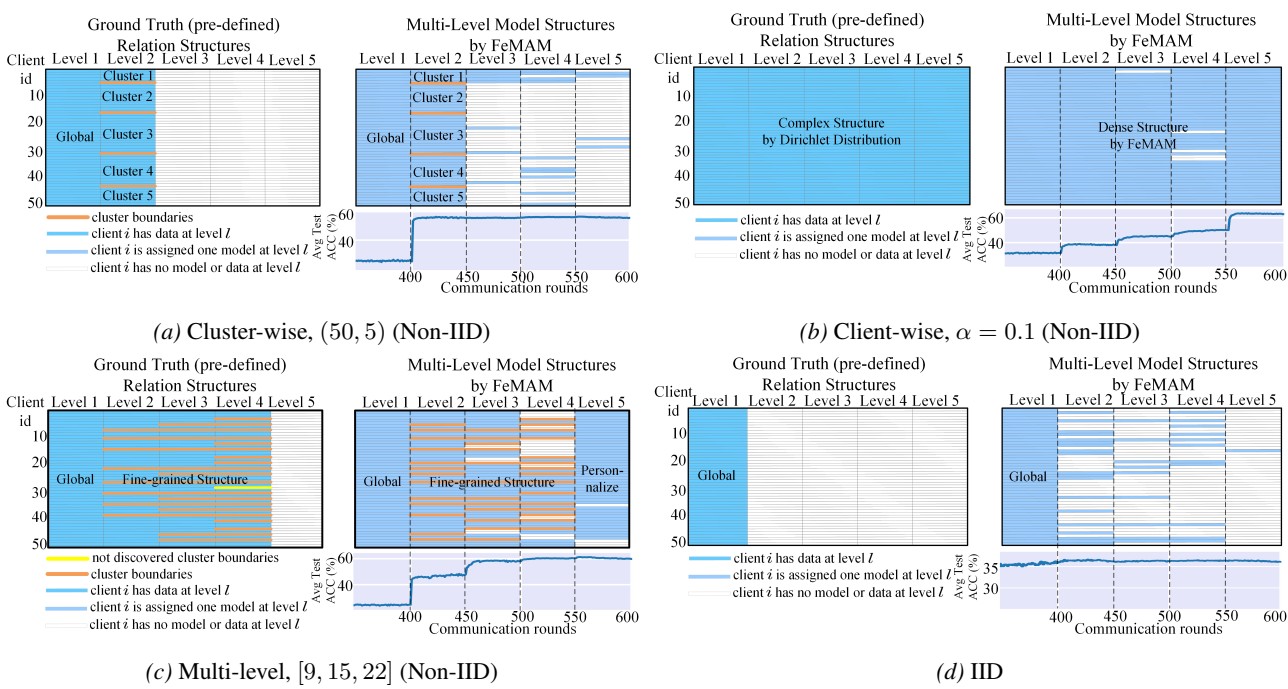

*Figure A.1.* FeMAM's learned multi-level structures (left), ground-truth (pre-defined) relation structures (right), and convergence curves (bottom) on Tiny ImageNet. Each learned structure contains 5 levels (columns) and 50 clients (rows). Cluster boundaries are marked when adjacent clients belong to different clusters (when applicable). Overall, FeMAM matches the major structural patterns while adaptively pruning levels that do not improve validation performance.

## B. Additional Experiments Details in Large-Scale, More Realistic Settings

**Model and PEFT configuration.** We adopt Qwen3-4B (Yang et al., 2025a) as the base model, with its thinking mode disabled. We insert LoRA adapters into the query and value projections. The LoRA rank is set to $r = 4$.

**Clients and non-IID construction via tasks.** We build a task-partitioned FL setting using tasks from LoraHub (Huang et al., 2023), which is derived from FLAN-T5 (Chung et al., 2024) and contains nearly 200 distinct tasks. For training efficiency, we select tasks with shorter sequence lengths (less than 64 tokens), resulting in **71 tasks**. We simulate an FL system with **71 clients**, where each client is assigned **one task** (i.e., one client $\leftrightarrow$ one task). For each client, the training/validation/test split contains **300/50/150** samples, respectively. The complete task list is provided in Table B.1.

**Baselines.** We consider the same set of baselines as in Section 5.1.1: Local, FedAvg, FedAvg+, Ditto, FeSEM, FedEM, pFedMoE, FedCAM, and FeMAM. We **exclude PM-MoE** because it is designed for classification-oriented settings (e.g., learning a router over classification heads) and relies on assumptions that do not align well with general sequence-to-sequence generation tasks and ROUGE-based evaluation in FLAN-style benchmarks.

We use the same hyperparameter settings as in Section 5.1.1 unless otherwise specified. In particular, FedEM, FedCAM, and FeSEM adopt 5 shared/global models. FeMAM follows the hierarchy in Table 2, except that at the personalized level we use **71** personal models (one per client/task).

**System settings and evaluation.** For all methods, we train for **100 communication rounds** and select the checkpoint with the smallest validation error. The number of local training epochs is set to 1. We report ROUGE-1 as the evaluation metric.

## C. Objective and Algorithm with FedAvg, K-means, and Local Train

Recall the overall objective:

$$\min_{\{r;\Theta\}} \; \mathcal{R}_L(\Theta^{(1:L)}, r^{(1:L)}) := \sum_{i=1}^{m} \frac{n_i}{n} \, \ell\left(Y_i, \; \sum_{l=1}^{L} \sum_{k=1}^{K_l} r_{i,k}^{(l)} f(X_i; \Theta_k^{(l)})\right). \tag{C.1}$$

The level-wise assignment rule is given by:

$$r_{i,k}^{(l)} \leftarrow \mathbb{I}\left[k = \arg\min_{k'} \mathcal{L}\left(D_i; \Theta_{k'}^{(l)}\right)\right], \tag{C.2}$$

where $\mathcal{L}$ denotes a general assignment criterion.

In Eq. (C.1), each level's model multiplicity $K_l$ and assignment criterion $\mathcal{L}$ need to be specified. As used in our experiments, Level 1 adopts FedAvg, Levels 2 to $(L-1)$ adopt K-means clustering, and Level $L$ adopts local training. Concretely, we set $K_1 = 1$ and $r_{i,1}^{(1)} = 1$. For $l = 2, \ldots, L-1$, we instantiate the assignment criterion in Eq. (C.2) as $\mathcal{L}(D_i; \Theta_k^{(l)}) := \|\theta_i^{(l)} - \Theta_k^{(l)}\|_2^2$, which yields the following instantiated objective:

$$\min_{\Theta,\theta,r} \sum_{i=1}^{m} \frac{n_i}{n} \left[ \ell\left(Y_i, \; f(X_i; \Theta_1^{(1)}) + \sum_{l=2}^{L-1} f(X_i; \theta_i^{(l)}) + f(X_i; \theta_i^{(L)})\right) + \frac{\lambda}{2} \sum_{l=2}^{L-1} \sum_{k=1}^{K_l} r_{i,k}^{(l)} \|\theta_i^{(l)} - \Theta_k^{(l)}\|_2^2 \right]$$

$$\text{s.t.} \quad r_{i,k}^{(l)} = \mathbb{I}\left[k = \arg\min_{k' \in [K_l]} \|\theta_i^{(l)} - \Theta_{k'}^{(l)}\|_2^2\right], \quad l = 2, \ldots, L-1, \; i = 1, \ldots, m, \tag{C.3}$$

$$r_{i,1}^{(1)} = 1.$$

Here, $\Theta$ denotes the shared models: a single global model $\Theta_1^{(1)}$ at Level 1 and cluster models $\{\Theta_k^{(l)}\}_{k=1}^{K_l}$ for $l = 2, \ldots, L-1$. $\theta$ denotes the client-side local models, including $\{\theta_i^{(l)}\}_{l=2}^{L}$, where Level $L$ is local. The instantiated FeMAM algorithm follows the procedure described in Algorithm 2.

*Table B.1.* Flan-T5 tasks used in our large-scale federated learning setting, grouped by task type. A total of 71 tasks are used, with each task assigned to one client.

| Task Type | Tasks |
|---|---|
| **Text Classification** | ag_news_subset:1.0.0, sentiment140:1.0.0, yelp_polarity_reviews:0.2.0, trec:1.0.0, glue_cola:2.0.0, glue_sst2:2.0.0, glue_qqp:2.0.0, glue_mnli:2.0.0, glue_qnli:2.0.0, glue_mrpc:2.0.0, glue_stsb:2.0.0, snli:1.1.0, paws_wiki:1.1.0, winogrande:1.1.0 |
| **Question Answering** | natural_questions_open:1.0.0, openbookqa:0.1.0, hellaswag:1.1.0, piqa:1.0.0, qasc_is_correct_1, qasc_is_correct_2, squad_v1.1:3.0.0, trivia_qa_rc:1.1.0, web_questions_get_the_answer, web_questions_potential_correct_answer, web_questions_question_answer, web_questions_short_general_knowledge_q, web_questions_whats_the_answer, wiki_qa_Is_This_True_, wiki_qa_found_on_google |
| **Reasoning / Multiple Choice** | ai2_arc_ARC-Challenge:1.0.0, ai2_arc_ARC-Easy:1.0.0, cos_e_v1.11_description_question_option_text, cos_e_v1.11_question_description_option_text, cos_e_v1.11_question_option_description_id, cos_e_v1.11_question_option_description_text |
| **Natural Language Generation** | gigaword:1.2.0, samsum:1.0.0, gem_common_gen:1.1.0, gem_dart:1.1.0, gem_e2e_nlg:1.1.0, gem_web_nlg_en:1.1.0, para_crawl_enes |
| **Reading Comprehension / Multi-hop QA** | kilt_tasks_hotpotqa_combining_facts, kilt_tasks_hotpotqa_complex_question, kilt_tasks_hotpotqa_final_exam, kilt_tasks_hotpotqa_formulate, kilt_tasks_hotpotqa_straighforward_qa |
| **Structured Prediction / Editing** | fix_punct, true_case, word_segment, definite_pronoun_resolution:1.1.0 |
| **Social Commonsense Reasoning** | social_i_qa_Check_if_a_random_answer_is_valid_or_not, social_i_qa_Generate_answer, social_i_qa_Generate_the_question_from_the_answer, social_i_qa_I_was_wondering, social_i_qa_Show_choices_and_generate_answer |
| **Math / Symbolic Reasoning** | math_dataset_algebra__linear_1d:1.0.0 |
| **Machine Translation** | wmt14_translate_fr-en:1.0.0, wmt16_translate_cs-en:1.0.0, wmt16_translate_de-en:1.0.0, wmt16_translate_fi-en:1.0.0, wmt16_translate_ro-en:1.0.0, wmt16_translate_ru-en:1.0.0, wmt16_translate_tr-en:1.0.0 |

# D. Convergence Analysis

## D.1. Level-wise objective and training protocol

Recall the overall objective:

$$\min_{\{r;\Theta\}} \mathcal{R}_L(\Theta^{(1:L)}, r^{(1:L)}) := \sum_{i=1}^{m} \frac{n_i}{n} \ell\left(Y_i, \sum_{l=1}^{L}\sum_{k=1}^{K_l} r_{i,k}^{(l)} f(X_i; \Theta_k^{(l)})\right). \tag{D.1}$$

The level-wise assignment rule is given by:

$$r_{i,k}^{(l)} \leftarrow \mathbb{I}\left[k = \arg\min_{k'} \mathcal{L}\left(D_i; \Theta_{k'}^{(l)}\right)\right], \tag{D.2}$$

where $\mathcal{L}$ denotes a general assignment criterion.

**Level-wise growing.** We define the truncated objective using only levels $1, \ldots, l$:

$$\mathcal{R}_l(\Theta^{(1:l)}, r^{(1:l)}) := \sum_{i=1}^{m} \frac{n_i}{n} \ell\left(Y_i, \sum_{l'=1}^{l}\sum_{k=1}^{K_{l'}} r_{i,k}^{(l')} f(X_i; \Theta_k^{(l')})\right), \qquad l = 0, 1, \ldots, L, \tag{D.3}$$

---

**Algorithm 2:** Federated Multi-Level Additive Modeling (FeMAM) instantiated with FedAvg, K-means, and Local Train

---

**Input** : Training & validation data, maximum level $L$.

**Output** : Models $\{\Theta_k^{(l)}\}$ and assignments $\{r_{i,k}^{(l)}\}$ for levels 1 to $L$.

1 **for** $l = 1$ **to** $L$ **do**
2     Initialize models $\{\Theta_k^{(l)}\}_{k=1}^{K_l}$ and assignments $\{r_{i,k}^{(l)}\}$;
3     **while** *not converge at level $l$* **do**
         /* Client updates (in parallel)                                        */
4         **foreach** *client $i$* **do**
5            Receive assigned model at level $l$ (and keep levels $< l$ frozen);
6            Update local model $\theta_i^{(l)}$ using Eq. (6);
7            **if** $l < L$ **then**
8               Upload $\theta_i^{(l)}$ to server;
         /* Server updates                                                  */
9         **if** $2 \leq l \leq L - 1$ **then**
10            Update assignments $\{r_{i,k}^{(l)}\}$ via Eq. (C.3);
11         **if** $l < L$ **then**
12            Aggregate level-$l$ models via Eq. (3);
13            Broadcast updated level-$l$ models to clients;
     /* Client-side structure pruning                                 */
14     **foreach** *client $i$* **do**
15         Accept level $l$ only if adding its output reduces validation error (Eq. (7));

---

with $\mathcal{R}_0 := \sum_{i=1}^{m} \frac{n_i}{n} \ell(Y_i, 0)$.

Training proceeds level by level. When training level $l$, all lower levels $(\Theta^{(1:l-1)}, r^{(1:l-1)})$ are frozen, and only $(\Theta^{(l)}, r^{(l)})$ are optimized for $T_l$ communication rounds.

**Notation.** For a fixed level $l$, define the frozen prediction from lower levels:

$$f_i^{(<l)}(X_i) := \sum_{l'=1}^{l-1} \sum_{k=1}^{K_{l'}} r_{i,k}^{(l')} f(X_i; \Theta_k^{(l')}),$$

and the level-$l$ additive contribution:

$$f_i^{(l)}(X_i) := \sum_{k=1}^{K_l} r_{i,k}^{(l)} f(X_i; \Theta_k^{(l)}).$$

Thus,

$$\mathcal{R}_l = \sum_{i=1}^{m} \frac{n_i}{n} \ell\Big(Y_i, \; f_i^{(<l)}(X_i) + f_i^{(l)}(X_i)\Big).$$

**Client-sum form and local-evaluated objective.** Fix level $l$ and assignments $r^{(l)}$, and define the client set of cluster $k$ as

$$c_l(k) := \{i : r_{i,k}^{(l)} = 1\}.$$

With lower levels frozen, the level-$l$ objective can be written explicitly as

$$\mathcal{R}_l(\Theta^{(l)}, r^{(l)}) = \sum_{k=1}^{K_l} \sum_{i \in c_l(k)} \frac{n_i}{n} \ell\Big(Y_i, \; f_i^{(<l)}(X_i) + f(X_i; \Theta_k^{(l)})\Big). \tag{D.4}$$

During one communication round $t$, each client $i \in c_l(k)$ maintains a local copy $\theta_{i,k}^{(t,q,l)}$ for $q = 0, \ldots, Q$, initialized by

$\theta_{i,k}^{(t,0,l)} = \Theta_k^{(t,l)}$. We also define the objective evaluated on these *parallel local copies*:

$$\mathcal{R}_l(\theta^{(t,q,l)}, r^{(l)}) := \sum_{k=1}^{K_l} \sum_{i \in c_l(k)} \frac{n_i}{n} \ell\left(Y_i,\ f_i^{(<l)}(X_i) + f(X_i; \theta_{i,k}^{(t,q,l)})\right). \tag{D.5}$$

Note that $\mathcal{R}_l(\theta^{(t,0,l)}, r^{(l)}) = \mathcal{R}_l(\Theta^{(t,l)}, r^{(l)})$.

## D.2. Assumptions

All assumptions below are stated for the *current level $l$*, since lower levels are frozen.

**Assumption D.1** (Unbiased stochastic gradients, bounded second moment, and bounded variance). For any client $i \in c_l(k)$ and any local step during level-$l$ training, the stochastic gradient $g_{i,k}^{(t,q,l)}$ used to update $\theta_{i,k}^{(t,q,l)}$ is unbiased:

$$\mathbb{E}[g_{i,k}^{(t,q,l)}] = \nabla \mathcal{R}_{l,i}(\theta_{i,k}^{(t,q,l)}).$$

Moreover, its second moment is bounded:

$$\mathbb{E}\|g_{i,k}^{(t,q,l)}\|_2^2 \le U^2,$$

and its variance is bounded:

$$\mathbb{E}\|g_{i,k}^{(t,q,l)} - \mathbb{E}g_{i,k}^{(t,q,l)}\|_2^2 \le \sigma^2.$$

**Assumption D.2** ($\beta$-smoothness). For fixed $(\Theta^{(1:l-1)}, r^{(1:l)})$, for any cluster $k$ and any client $i \in c_l(k)$, the function $\theta \mapsto \mathcal{R}_{l,i}(\theta)$ is $\beta$-smooth. That is, for all $\Delta$,

$$\mathcal{R}_{l,i}(\Theta_k^{(l)} + \Delta) \le \mathcal{R}_{l,i}(\Theta_k^{(l)}) + \left\langle \nabla \mathcal{R}_{l,i}(\Theta_k^{(l)}),\ \Delta \right\rangle + \frac{\beta}{2}\|\Delta\|_2^2.$$

**Assumption D.3** (Clusterability / gradient diversity within a cluster). For any cluster $k \in [K_l]$, define within-cluster weights

$$w_{i|k} := \frac{n_i}{\sum_{p \in c_l(k)} n_p}, \qquad \bar{g}_k^{(t,q,l)} := \sum_{p \in c_l(k)} w_{p|k}\, g_{p,k}^{(t,q,l)}.$$

There exists a constant $B_l \ge 0$ such that for all $t, q, k$ and all $i \in c_l(k)$,

$$\|g_{i,k}^{(t,q,l)} - \bar{g}_k^{(t,q,l)}\|_2 \le B_l \|\bar{g}_k^{(t,q,l)}\|_2.$$

**Assumption D.4** (Abstract assignment update with bounded slack). The assignment update rule (D.2) is treated as an abstract assignment operator. Although $\mathcal{L}$ may not coincide with the true objective $\mathcal{R}_l$, we assume that its effect on the objective is bounded as follows: for each assignment update at level $l$,

$$\mathcal{R}_l(\Theta^{(l)}, r^{(l,+)}) \le \mathcal{R}_l(\Theta^{(l)}, r^{(l)}) + \varepsilon_l, \tag{D.6}$$

where $r^{(l,+)}$ denotes the updated assignments and $\varepsilon_l \ge 0$ is a level-dependent constant.

**Discussion.** Assumption D.4 abstracts a broad class of level-wise assignment mechanisms induced by $\mathcal{L}$ in (D.2). For IFCA-style routing, assignments are obtained by directly minimizing the loss criterion $\mathcal{L}(D_i; \Theta_k^{(l)})$ over candidate models, and thus introduce no additional mismatch, yielding $\varepsilon_l = 0$. In contrast, K-means clustering assigns clients via a distance-to-centroid surrogate (e.g., $\|\theta_i^{(l)} - \Theta_k^{(l)}\|_2$), which differ from loss-based optimal routing; in this case, the induced slack $\varepsilon_l$ can be bounded in terms of within-cluster dispersion as in (Ma et al., 2022).

**Assumption D.5** (Boosting edge / residual alignment). Let

$$g^{(<l)} := -\nabla_{f^{(l)}} \mathcal{R}_l$$

denote the functional descent direction induced by lower levels. We assume that after training level $l$, there exists $\gamma_l \in (0, 1]$ such that

$$\left\langle g^{(<l)}, h^{(l)} \right\rangle \ge \gamma_l \|g^{(<l)}\|_2 \|h^{(l)}\|_2.$$

i.e., the level-$l$ additive predictor is sufficiently aligned with the residual.

## D.3. One-round bounds at a fixed level

Fix a level $l$ and consider communication rounds $t = 0, \ldots, T_l - 1$. Define

$$\mathcal{R}_l^{(t)} := \mathcal{R}_l(\Theta^{(t,l)}, r^{(l)}).$$

### D.3.1. LOCAL UPDATE BOUND

**Lemma D.6** (Local SGD descent at level $l$ (client-sum form)). *Under Assumptions D.1 and D.2, if $\eta_l \in (0, 1/\beta]$, then*

$$\mathbb{E}\Big[\mathcal{R}_l(\theta^{(t,Q,l)}, r^{(l)}) - \mathcal{R}_l(\theta^{(t,0,l)}, r^{(l)})\Big] \leq -\Big(\eta_l - \frac{\beta\eta_l^2}{2}\Big) \sum_{q=0}^{Q-1} \sum_{k=1}^{K_l} \sum_{i \in c_l(k)} \frac{n_i}{n} \mathbb{E}\Big\|\nabla\mathcal{R}_{l,i}(\theta_{i,k}^{(t,q,l)})\Big\|_2^2 + \frac{\beta\eta_l^2}{2} Q\sigma^2. \quad \text{(D.7)}$$

**Proof (routine).** Fix any client $i \in c_l(k)$ and one local step $\theta_{i,k}^{(t,q+1,l)} = \theta_{i,k}^{(t,q,l)} - \eta_l g_{i,k}^{(t,q,l)}$. By $\beta$-smoothness of $\mathcal{R}_{l,i}$ (Assumption D.2),

$$\mathcal{R}_{l,i}(\theta_{i,k}^{(t,q+1,l)}) \leq \mathcal{R}_{l,i}(\theta_{i,k}^{(t,q,l)}) - \eta_l \langle \nabla\mathcal{R}_{l,i}, g_{i,k}^{(t,q,l)} \rangle + \frac{\beta\eta_l^2}{2} \|g_{i,k}^{(t,q,l)}\|_2^2.$$

Taking expectation and using unbiasedness and variance bound (Assumption D.1) yields

$$\mathbb{E}\langle \nabla\mathcal{R}_{l,i}, g_{i,k} \rangle = \|\nabla\mathcal{R}_{l,i}\|_2^2, \qquad \mathbb{E}\|g_{i,k}\|_2^2 \leq \|\nabla\mathcal{R}_{l,i}\|_2^2 + \sigma^2,$$

hence

$$\mathbb{E}\big[\mathcal{R}_{l,i}(\theta_{i,k}^{(t,q+1,l)}) - \mathcal{R}_{l,i}(\theta_{i,k}^{(t,q,l)})\big] \leq -\Big(\eta_l - \frac{\beta\eta_l^2}{2}\Big) \mathbb{E}\|\nabla\mathcal{R}_{l,i}\|_2^2 + \frac{\beta\eta_l^2}{2}\sigma^2.$$

Summing over $q = 0, \ldots, Q - 1$, then multiplying by $\frac{n_i}{n}$ and summing over all clients $i$ yields (D.7) via (D.5). $\square$

### D.3.2. AGGREGATION BOUND (NONCONVEX)

**Lemma D.7** (Aggregation drift at level $l$ (nonconvex, via smoothness)). *Under Assumptions D.1, D.2, and D.3, the aggregation step satisfies*

$$\mathbb{E}\Big[\mathcal{R}_l(\Theta^{(t+1,l)}, r^{(l)}) - \mathcal{R}_l(\theta^{(t,Q,l)}, r^{(l)})\Big] \leq \eta_l B_l Q U^2 + \frac{\beta}{2}\eta_l^2 B_l^2 Q^2 U^2. \quad \text{(D.8)}$$

**Proof (routine).** For each cluster $k$, the server aggregates

$$\Theta_k^{(t+1,l)} = \sum_{i \in c_l(k)} w_{i|k} \theta_{i,k}^{(t,Q,l)}, \qquad w_{i|k} := \frac{n_i}{\sum_{p \in c_l(k)} n_p}.$$

Define $\delta_{i,k}^{(t)} := \Theta_k^{(t+1,l)} - \theta_{i,k}^{(t,Q,l)}$. By the local update recursion,

$$\theta_{i,k}^{(t,Q,l)} = \Theta_k^{(t,l)} - \eta_l \sum_{q=0}^{Q-1} g_{i,k}^{(t,q,l)}, \quad \Theta_k^{(t+1,l)} = \Theta_k^{(t,l)} - \eta_l \sum_{q=0}^{Q-1} \bar{g}_k^{(t,q,l)},$$

hence

$$\delta_{i,k}^{(t)} = \eta_l \sum_{q=0}^{Q-1} \big(g_{i,k}^{(t,q,l)} - \bar{g}_k^{(t,q,l)}\big).$$

Using Assumption D.3 and triangle inequality,

$$\|\delta_{i,k}^{(t)}\|_2 \leq \eta_l \sum_{q=0}^{Q-1} \|g_{i,k}^{(t,q,l)} - \bar{g}_k^{(t,q,l)}\|_2 \leq \eta_l B_l \sum_{q=0}^{Q-1} \|\bar{g}_k^{(t,q,l)}\|_2.$$

Moreover, by Jensen and Assumption D.1,

$$\mathbb{E}\|\bar{g}_k^{(t,q,l)}\|_2^2 = \mathbb{E}\Big\|\sum_{i \in c_l(k)} w_{i|k} g_{i,k}^{(t,q,l)}\Big\|_2^2 \leq \sum_{i \in c_l(k)} w_{i|k}\, \mathbb{E}\|g_{i,k}^{(t,q,l)}\|_2^2 \leq U^2.$$

Thus, by Cauchy–Schwarz,

$$\mathbb{E}\|\delta_{i,k}^{(t)}\|_2 \leq \eta_l B_l \sum_{q=0}^{Q-1} \sqrt{\mathbb{E}\|\bar{g}_k^{(t,q,l)}\|_2^2} \leq \eta_l B_l Q U, \qquad \mathbb{E}\|\delta_{i,k}^{(t)}\|_2^2 \leq (\eta_l B_l Q)^2 U^2.$$

Now apply $\beta$-smoothness (Assumption D.2) to each client loss: for $i \in c_l(k)$,

$$\mathcal{R}_{l,i}(\Theta_k^{(t+1,l)}) \leq \mathcal{R}_{l,i}(\theta_{i,k}^{(t,Q,l)}) + \left\langle \nabla\mathcal{R}_{l,i}(\theta_{i,k}^{(t,Q,l)}),\, \delta_{i,k}^{(t)} \right\rangle + \frac{\beta}{2}\|\delta_{i,k}^{(t)}\|_2^2.$$

Multiply by $\frac{n_i}{n}$ and sum over all $k$ and $i \in c_l(k)$, yielding

$$\mathcal{R}_l(\Theta^{(t+1,l)}, r^{(l)}) - \mathcal{R}_l(\theta^{(t,Q,l)}, r^{(l)}) \leq \sum_k \sum_{i \in c_l(k)} \frac{n_i}{n} \left\langle \nabla\mathcal{R}_{l,i}(\theta_{i,k}^{(t,Q,l)}),\, \delta_{i,k}^{(t)} \right\rangle + \frac{\beta}{2} \sum_k \sum_{i \in c_l(k)} \frac{n_i}{n}\|\delta_{i,k}^{(t)}\|_2^2.$$

Taking expectation and using Cauchy–Schwarz together with $\mathbb{E}\|\nabla\mathcal{R}_{l,i}(\theta)\|_2^2 \leq \mathbb{E}\|g\|_2^2 \leq U^2$ from Assumption D.1, we obtain

$$\mathbb{E}\left\langle \nabla\mathcal{R}_{l,i}(\theta_{i,k}^{(t,Q,l)}),\, \delta_{i,k}^{(t)} \right\rangle \leq \sqrt{\mathbb{E}\|\nabla\mathcal{R}_{l,i}(\theta_{i,k}^{(t,Q,l)})\|_2^2} \cdot \sqrt{\mathbb{E}\|\delta_{i,k}^{(t)}\|_2^2} \leq U \cdot (\eta_l B_l Q U).$$

Since $\sum_k \sum_{i \in c_l(k)} \frac{n_i}{n} = 1$, we conclude (D.8). $\qquad\qquad\square$

### D.3.3. PER-ROUND BOUND

**Theorem D.8** (One-round descent at level $l$). *Combining Lemmas D.6, D.7, and the assignment slack (D.6), we have*

$$\mathbb{E}[\mathcal{R}_l^{(t+1)}] \leq \mathbb{E}[\mathcal{R}_l^{(t)}] + \varepsilon_l - \left(\eta_l - \frac{\beta\eta_l^2}{2}\right)\sum_{q=0}^{Q-1}\sum_{k=1}^{K_l}\sum_{i \in c_l(k)} \frac{n_i}{n}\, \mathbb{E}\Big\|\nabla\mathcal{R}_{l,i}(\theta_{i,k}^{(t,q,l)})\Big\|_2^2 + \frac{\beta\eta_l^2}{2}Q\sigma^2 + \eta_l B_l Q U^2 + \frac{\beta}{2}\eta_l^2 B_l^2 Q^2 U^2.$$
$$\text{(D.9)}$$

### D.4. Telescoping over rounds at level $l$

Summing (D.9) over $t = 0, \ldots, T_l - 1$ yields

$$\left(\eta_l - \frac{\beta\eta_l^2}{2}\right)\sum_{t=0}^{T_l-1}\sum_{q=0}^{Q-1}\sum_{k=1}^{K_l}\sum_{i \in c_l(k)} \frac{n_i}{n}\, \mathbb{E}\Big\|\nabla\mathcal{R}_{l,i}(\theta_{i,k}^{(t,q,l)})\Big\|_2^2 \leq \Delta_l + T_l\varepsilon_l + T_l\left(\frac{\beta\eta_l^2}{2}Q\sigma^2 + \eta_l B_l Q U^2 + \frac{\beta}{2}\eta_l^2 B_l^2 Q^2 U^2\right),$$
$$\text{(D.10)}$$

where $\Delta_l := \mathcal{R}_l^{(0)} - \inf \mathcal{R}_l$.

**Step-size regime.** Throughout, we require

$$0 < \eta_l \leq \frac{1}{\beta}. \qquad\qquad\qquad\qquad \text{(D.11)}$$

**Within-level averaged stationarity.** Define $\alpha_l := \eta_l - \frac{\beta\eta_l^2}{2} > 0$. Dividing (D.10) by $T_l Q\, \alpha_l$ gives

$$\frac{1}{T_l Q}\sum_{t=0}^{T_l-1}\sum_{q=0}^{Q-1}\sum_{k=1}^{K_l}\sum_{i \in c_l(k)} \frac{n_i}{n}\, \mathbb{E}\Big\|\nabla\mathcal{R}_{l,i}(\theta_{i,k}^{(t,q,l)})\Big\|_2^2 \leq \frac{\Delta_l}{T_l Q\, \alpha_l} + \frac{\varepsilon_l}{Q\, \alpha_l} + \frac{\frac{\beta\eta_l^2}{2}\sigma^2 + \eta_l B_l U^2 + \frac{\beta}{2}\eta_l^2 B_l^2 Q U^2}{\alpha_l}. \qquad \text{(D.12)}$$

### D.5. Effect of adding a new level

We now show that after completing training at level $l$, the objective improves due to two mechanisms.

**(i) Structure pruning.** After training level $l$, client $i$ accepts the new level only if

$$\ell(Y_i, f_i^{(<l)}(X_i)) > \ell(Y_i, f_i^{(<l)}(X_i) + f_i^{(l)}(X_i)). \tag{D.13}$$

Otherwise, the assignment is removed (equivalently, set $r_{i,k}^{(l)} = 0$ for all $k$). Thus, for each client, the accepted level-$l$ contribution never increases its loss, and therefore

$$\mathcal{R}_l \le \mathcal{R}_{l-1}. \tag{D.14}$$

**(ii) Boosting improvement.** Let $F^{(<l)} \in \mathbb{R}^m$ denote the vector of lower-level predictions $F_i^{(<l)} := f_i^{(<l)}(X_i)$, and let $h^{(l)} \in \mathbb{R}^m$ denote the level-$l$ additive outputs $h_i^{(l)} := f_i^{(l)}(X_i)$. By $\beta$-smoothness w.r.t. the prediction vector, for any $\eta > 0$,

$$\mathcal{R}_{l-1}(F^{(<l)} + \eta h^{(l)}) \le \mathcal{R}_{l-1}(F^{(<l)}) + \eta \langle \nabla \mathcal{R}_{l-1}(F^{(<l)}), h^{(l)} \rangle + \frac{\beta}{2} \eta^2 \|h^{(l)}\|_2^2. \tag{D.15}$$

Let

$$g^{(<l)} := -\nabla \mathcal{R}_{l-1}(F^{(<l)})$$

be the residual (negative gradient) with respect to the prediction vector. Then $\langle \nabla \mathcal{R}_{l-1}, h^{(l)} \rangle = -\langle g^{(<l)}, h^{(l)} \rangle$ and (D.15) becomes

$$\mathcal{R}_{l-1}(F^{(<l)} + \eta h^{(l)}) \le \mathcal{R}_{l-1}(F^{(<l)}) - \eta \langle g^{(<l)}, h^{(l)} \rangle + \frac{\beta}{2} \eta^2 \|h^{(l)}\|_2^2.$$

Under Assumption D.5, $\langle g^{(<l)}, h^{(l)} \rangle \ge \gamma_l \|g^{(<l)}\|_2 \|h^{(l)}\|_2$. Choosing the (upper-bound optimal) step-size

$$\frac{\langle g^{(<l)}, h^{(l)} \rangle}{\beta \|h^{(l)}\|_2^2} \ge \frac{\gamma_l}{\beta} \frac{\|g^{(<l)}\|_2}{\|h^{(l)}\|_2},$$

and substituting into (D.15) yields the standard boosting decrease:

$$\mathcal{R}_l \le \mathcal{R}_{l-1} - \frac{\langle g^{(<l)}, h^{(l)} \rangle^2}{2\beta \|h^{(l)}\|_2^2} \le \mathcal{R}_{l-1} - \frac{\gamma_l^2}{2\beta} \|g^{(<l)}\|_2^2 = \mathcal{R}_{l-1} - \frac{\gamma_l^2}{2\beta} \|\nabla \mathcal{R}_{l-1}\|_2^2. \tag{D.16}$$

**Combined effect.** Structure pruning guarantees the non-increase (D.14), while the boosting edge yields the strict descent (D.16) whenever the residual is non-zero. Hence, after finishing level $l$ training,

$$\mathcal{R}_l \le \mathcal{R}_{l-1} - \frac{\gamma_l^2}{2\beta} \|\nabla \mathcal{R}_{l-1}\|_2^2. \tag{D.17}$$

### D.6. Telescoping over levels and final convergence statement

We have established two complementary properties: (i) **within-level convergence** for each fixed level $l$ via (D.10)–(D.12), and (ii) **across-level descent** when adding a new level via (D.17). We now summarize their implications.

**Across-level monotone descent.** Equation (D.17) implies the objective is monotone non-increasing across levels:

$$\mathcal{R}_0 \ge \mathcal{R}_1 \ge \cdots \ge \mathcal{R}_L.$$

Summing (D.17) over $l = 1, \ldots, L$ yields

$$\mathcal{R}_L \le \mathcal{R}_0 - \sum_{l=1}^{L} \frac{\gamma_l^2}{2\beta} \|\nabla \mathcal{R}_{l-1}\|_2^2. \tag{D.18}$$

**Averaged-stationarity over all levels.** Let $N_{\text{tot}} := \sum_{l=1}^{L} T_l Q$ and $\alpha_l := \eta_l - \frac{\beta \eta_l^2}{2}$. Summing (D.10) over $l = 1, \ldots, L$ and dividing by $N_{\text{tot}}$ gives

$$\frac{1}{N_{\text{tot}}} \sum_{l=1}^{L} \sum_{t=0}^{T_l-1} \sum_{q=0}^{Q-1} \sum_{k=1}^{K_l} \sum_{i \in c_l(k)} \frac{n_i}{n} \mathbb{E} \left\| \nabla \mathcal{R}_{l,i}(\theta_{i,k}^{(t,q,l)}) \right\|_2^2 \leq \frac{\sum_{l=1}^{L} \frac{1}{\alpha_l} \left( \Delta_l + T_l \varepsilon_l + T_l \left( \frac{\beta \eta_l^2}{2} Q \sigma^2 + \eta_l B_l Q U^2 + \frac{\beta}{2} \eta_l^2 B_l^2 Q^2 U^2 \right) \right)}{\sum_{l=1}^{L} T_l Q}. \tag{D.19}$$

**Conclusion.** Equations (D.18) and (D.12)–(D.19) together show: (i) adding levels yields a monotone non-increasing objective sequence $\{\mathcal{R}_l\}_{l=0}^{L}$, and (ii) optimization at each level (and in aggregate across all levels) satisfies a standard averaged-gradient convergence bound.

