# OpenReview forum: "Personalized Additive Modeling for Multi-level Federated Learning"
_ICML.cc/2026/Conference — ICML 2026 regular_

### Official Review · Reviewer_xDZT · 2026-03-11

**Soundness:** 3
**Presentation:** 4
**Significance:** 3
**Originality:** 4
**Overall Recommendation:** 5
**Confidence:** 4

**Summary:**

This paper proposes FedMAM, a multi-level Personalized Federated Learning framework that addresses various non-IID scenarios within a unified framework. Specifically, each client-specific personalized model is represented as a unique additive structure composed of multiple model blocks across different levels.

**Compliance With Llm Reviewing Policy:**

Affirmed.

**Final Justification:**

My concerns have been fully addressed. After reviewing the other comments and responses, I have no further questions.

**Key Questions For Authors:**

Please refer to the questions listed in the Weaknesses section.
Additional minor questions are listed below.
1. In Appendix A, could you provide practical guidelines for selecting KKK across different levels and for cluster initialization?

2. In Appendix D, could you clarify what types of tasks are introduced in LoRAHub? The appendix should ideally be self-contained.

**Limitations:**

It is suggested that the author add a discussion of limitations and future work.

**Strengths And Weaknesses:**

*** Strengths

1. The targeted problem is important. In Federated Learning (FL), non-IID data across clients is a key challenge that distinguishes FL from classical machine learning, which usually assumes that samples are drawn from an IID dataset or distribution. Although many personalized FL methods have been proposed, they are typically designed for specific hidden relational structures of non-IID data among clients. This paper attempts to propose a learning framework that can unify multiple non-IID learning scenarios with different relational structures.

2. The architecture design is novel and technically sound. The motivation behind the architecture design is clearly explained using a stacked residual block architecture (Figure 2). The proposed multi-level additive mechanism formulates client-specific personalized FL through a multi-level structure (Figure 3). The learning objective and algorithmic implementation are well supported by the theoretical analysis.

3. The claims are supported by comprehensive experimental analysis. Figure 4 visualizes that the proposed FedMAM can effectively capture various hidden relational structures among non-IID datasets across clients. In Table 1, the proposed method outperforms all baselines across various non-IID scenarios, including client-wise, cluster-wise, and multi-level relational structures.

4. Reproducibility is well supported. The reproducibility of the work is strengthened by the detailed experimental settings and comprehensive analysis provided in the appendix. Moreover, the source code is provided as a ZIP file.


*** Weaknesses

1. Scalability of the framework is unclear. The proposed framework appears to be a strong baseline that can handle various scenarios. However, it is unclear how the method scales to a large multi-level FL framework. The current experiments evaluate up to five levels. I am interested in understanding the capacity boundary of this framework. Are there more complex scenarios that require stronger modeling capacity? In such cases, the framework might need to scale to hundreds of levels, similar to large language models. Although the paper briefly mentions this as future work, it would be helpful to clarify whether the proposed multi-level framework can scale simply by stacking more levels or by increasing the number of clusters.

2. Integration of structure pruning is unclear. Although structure pruning is introduced as an effective mechanism to improve efficiency, it is not clear how it is integrated into the learning procedure described in Algorithm 1. For example, is pruning applied only after the learning procedure is completed, or is it performed during training?

3. Generalization to unseen clients is not discussed. Most personalized FL methods perform well for clients that participate in training. Could the authors discuss the framework’s generalization capability for new clients that are unseen during the training stage?

---

> ### Author Rebuttal · Authors · 2026-03-28
>
> ### W1. The scalability of FeMAM to larger and more complex multi-level FL scenarios should be clarified.
>
> Thanks for your interest in the potential scalability of our work. Complex scenarios that require multiple models have been validated in recent years. As a rough example, S-LoRA [1] shows that a shared base model can serve thousands of concurrent adapters with small overhead; Also, model merging [2] results show that multiple specialist models can be merged into one capable model; These examples suggest that recent multi-model systems are to ensemble many specialized models or adapters.
>
> Due to FeMAM’s unique design, we believe FeMAM  is able to significantly outperform previous methods in large-scale settings. Previous fixed-level FL methods (e.g., personalized FL or clustered FL) rely on inflexible predefined assumptions about non-IIDness, which often mismatch real-world complexity and thus lead to failure. In contrast, FeMAM does not impose strong prior assumptions on the data distribution. It starts from a basic model space and progressively tries to expand the model space via boosting-like optimization, and only keeps good models while ignoring bad ones via structure pruning. As a result, FeMAM is able to continually explore more possible non-IID patterns by increasing the number of models (i.e., larger L and K).
>
> [1] Sheng Y, Cao S, Li D, et al. S-lora: Serving thousands of concurrent lora adapters[J]. arXiv preprint arXiv:2311.03285, 2023.
>
> [2] Yang E, Shen L, Guo G, et al. Model merging in llms, mllms, and beyond: Methods, theories, applications, and opportunities[J]. ACM Computing Surveys, 2026, 58(8): 1-41.
>
> ### W2. The integration of structure pruning into the learning procedure should be clarified.
>
> For each client, the contribution of a new model cannot be determined only after training is completed. Therefore, at each level, all clients participate in training the new models. Once training at each level is completed, the trained new models are sent to each client for validation. Each client only keeps the new models that reduce its validation error.
>
> ### W3. The generalization capability of FeMAM to unseen clients should be discussed.
>
> For new clients, the existing FeMAM model structure remains valid as long as their data distribution lies within the distributions of existing clients. Thus, new clients can directly reuse the existing models, i.e., sequentially send levels of existing models to new clients, while news clients only keep useful models. The performance of new clients are expected to be close to existing clients with similar data distributions.
>
> For clients with unseen data distributions, our framework does not provide a solution. However, new distributions can still share similarities with existing ones. A practical approach is to finetune existing models, i.e., sequentially send the level-wise models to the new client, fine-tunes these models one by one and retains only the useful models.
>
> ### Q1. Practical guidelines for selecting KKK across levels and for cluster initialization should be provided.
>
> Selection of K. As shown in Table A.1 in Appendix A, FeMAM is not sensitive to the choice of K. As long as K lies within a reasonable range (1 < K < m) and multiple clustered levels are available, FeMAM has sufficient flexibility to approximate the underlying structure. As a practical guideline, since FeMAM adopts K-means clustering (FeSEM), K can be determined using standard methods for K-means, e.g., Elbow or Silhouette criteria. Nevertheless, such strategies are not strict, and FeMAM is robust to different choices of KKK without relying on them.
>
> Cluster initialization. At each level, the model fits the residual between the outputs of previous levels and the ground truth. Because the residuals differ across levels, both the cluster assignments and the corresponding models need to be randomly initialized at each level.
>
> ### Q2. The task types introduced in LoRAHub should be clarified so that Appendix D is more self-contained.
>
> The dataset used in LoRAHub[1] is derived from FLAN-T5[2], converging around 200 natural language instruction-following tasks. These tasks cover common categories in FLAN-T5, including question answering, reading comprehension / reasoning,  classification, knowledge / fact QA, generation and commonsense reasoning. We provide a complete list of the subset of LoRAHub tasks used in our experiments, as detailed in Table D.1 in Appendix D. We will make this clearer in the future.
>
> [1] Huang C, Liu Q, Lin B Y, et al. Lorahub: Efficient cross-task generalization via dynamic lora composition[J]. arXiv preprint arXiv:2307.13269, 2023.
>
> [2]Chung H W, Hou L, Longpre S, et al. Scaling instruction-finetuned language models[J]. Journal of Machine Learning Research, 2024, 25(70): 1-53.

---

> > ### Author Rebuttal · Reviewer_xDZT · 2026-04-02
> >
> > Thank you for your response. My concerns have been fully addressed. After reviewing the other comments and responses, I have no further questions. I am willing to increase my score to 5.

---

### Official Review · Reviewer_e3kr · 2026-03-11

**Soundness:** 2
**Presentation:** 3
**Significance:** 2
**Originality:** 2
**Overall Recommendation:** 3
**Confidence:** 4

**Summary:**

The paper introduces FeMAM algorithm which is a multi level federated learning framework. The key idea is FeMAM represent each client's personalized predictor as an additive composition of models drawn from multiple hierarchy levels. Experiments on CIFAR10 andTiny imagenet datasets show that FeMAM is outperforms personalized FL baselines.

**Compliance With Llm Reviewing Policy:**

Affirmed.

**Key Questions For Authors:**

**Check the Weaknesses.**

**Q1-** How does FeMAM perform under partical participation for example when some clients skip rounds ? I am woundering if the idea of freeze previuse levels will help or hurt FeMAM at this case.

 **Q2-** In the foundation model experiment in Lora setting, how is the additive composition implmented across levels ?

**Limitations:**

Yes

**Strengths And Weaknesses:**

**Strengths:**

* The paper is well written and easy to follow.
* The paper introduce a practical training protocol where one level is active per round and the others frozen. This is indeed a clean way to control per round cost and stabilize optimization.
* The authors show intesive evaluetion of FeMAM on two different dataset CIFAR100 and TinyImage across different non-iid partitions
* FeMAM shows also gains in a task partitioned foundation model setting using Qwen3-4b.

**Weaknesses:**
* Acodring to my understanding of how FeMAM work, it first do effective Fedavg and then repeated k-means clustered Fedeared learning levels and then local training. This framwork can be seen as composing existing building blocks rather than introducing a fundmanetally new optimixation principle. FedAvg is a standard baseline, Clustered FL with Kmeans is standard as well ( such as FeSEM [1]  which then limit the novelty of FeMAM.

* In The theory part, The key monotonicity claim in the main paper is presented as an intution only and later on in the appendix the intorduced detailed theory relies on multiple strong assumptions. For example the assumption of validation based pruning and the residual alignment assumption $\langle g^{(<l)}, h^{(l)} \rangle \ge \gamma_l \|g^{(<l)}\|_2 \|h^{(l)}\|_2$ are non-standard strong assumptions that  leads to non-standard converge result for nonconvecx deep FL.

*  FeMAM  personalized predictor is defined as a sum of the level wise model outputs  $ f_i(x) = \sum_{l,k} r_{i,k}^{(l)} f(x; {\Theta_{k}^{(l)}}) $ it's unclear to me and i can't find the answer in the paper, what object is being added in classification ? row logits ? probabilites ? something else ? I belive this ambiguity matters a lot because if FeMAM sums logits then the final prediction is a logit ensemble before softmax right ? and if FeMAM sums probabilities then the loss and the initial properties change. Also how about the foundation model epxeriments as well ? it is even less obvious wherer FeMAM adding token logits , adapter outputs or full model predictions ?

* One of my main concerns here regarding the practical use of this method is the inference cost. The authors clearly state that a client may store up to $L$ models and that inference can be up to an $L$ fold compute increase compare to a single model method. They also reported that FeMAM retains an average of $4.7$ models per client in both the client wise and multi level non-iid case which is very close to their maximum depth $L=5$ that typically leads to substantially higher storage and compute at deployment.


**References:**

[1] Long, Guodong, et al. "Multi-center federated learning: clients clustering for better personalization"

---

> ### Author Rebuttal · Authors · 2026-03-28
>
> ### W1. FeMAM mainly composes standard FL building blocks, so its novelty beyond FedAvg, clustered FL, and local training needs clarification.
>
> The contribution of FeMAM is to build a general FL framework for complex multi-level non-IID scenarios, a previously unexplored and challenging setting. To achieve this goal, we model multi-level non-IID into a novel architecture by including server novel designs, as stated in the methodology: additive modeling for integrating different levels of models, boosting-like optimization, and structure pruning for efficient optimization. All these novel designs lead to a new FeMAM algorithm and a new optimization principle, as shown in Algorithm 1. The principal design of FeMAM is general and not limited to specific level-wise components, e.g., FedAvg or FeSEM (K-means).
>
> FedAvg and FeSEM (K-means) are benchmark FL methods for solving global IID or clustering-based non-IID problems in FL, but they are not designed for multi-level non-IID scenarios. In our experiments, these benchmark methods are utilized as level-wise components within the FeMAM framework due to their representativeness and generality, and FeMAM successfully absorbs their abilities to serve a more general multi-level non-IID objective. Without the novel designs in FeMAM, these benchmark methods alone are insufficient.
>
> ### W2. The theoretical monotonicity claim relies on strong assumptions, so the role and reasonableness of these assumptions need clarification.
>
> In gradient boosting, monotonic improvement relies on an alignment condition between the learned component and the residual (i.e., a weak learning assumption). Such alignment is not guaranteed with deep neural networks in non-convex settings, and end-to-end convergence remains largely an open challenge. Therefore, a typical approach is to treat this alignment as an analysis condition rather than a guaranteed property, supported by the empirical effectiveness of widely used deep neural network–based boosting methods. Similarly, we adopt an alignment condition in our non-convex and federated setting to characterize the descent behavior. Indeed, we agree while such analysis is typical, it remains conditional in non-convex deep learning. We will state such analysis more clearly in the appendix. In addition, validation-based pruning is not an assumption, but an explicit algorithmic design that must be considered in the theoretical analysis.
>
> ### W3. The additive predictor is ambiguous, so clarification is needed on what is actually being summed in classification and LoRA-based foundation model experiments.
>
> FeMAM for classification tasks: The outputs being summed are the logits from different level-wise models. The aggregated logits are then passed through a softmax for prediction.
>
> $$
> h=\mathrm{softmax}\left(\sum_{i=1}^{K} f_i(x)\right)
> $$
>
> FeMAM with LoRA: The outputs of multiple LoRA adapters are summed. Formally, for a layer with weight $W$, LoRA models the output as:
>
> $$
> h = x \left( W + \sum_{i=1}^{K} B_i A_i \right)
> = xW + \sum_{i=1}^{K} x B_i A_i
> $$
>
> Thanks for pointing this out. We will make these details clearer in the paper.
>
> ### W4. FeMAM's inference cost may be high (retaining 4.7 models per client on average in multi-level and client-wise non-IID settings).
>
> $L$ is the maximum number of models that can be assigned to a client. As discussed in the Cost Analysis section, for clients with limited capacity, FeMAM can adaptively allocate fewer models. Therefore, the inference cost of FeMAM does not exceed the capacity of each client.
>
> Moreover, in multi-level non-IID and client-wise non-IID settings, although the compared methods use fewer models, their performance is significantly worse than FeMAM. FeMAM employs more models because it automatically captures the increased structural complexity of these scenarios and allocates additional models accordingly to achieve better performance, whereas the baselines rely on a fixed number of models. For simpler scenarios (e.g., cluster-wise non-IID and IID), FeMAM requires only a small number of models to outperform the baselines.
>
> ### Q1. FeMAM with partial participation, and whether freezing previous levels would help or hurt.
>
> We provide results by randomly sampling clients at each round on the client-wise non-IID CIFAR-100 dataset:
>
> |Client Partition Ratio|Acc|
> |:-:|-|
> |0.1|58.90|
> |0.5|61.72|
> |0.9|62.06|
> |1.0|61.96|
>
> Skipping some clients does not lead to a drastic performance decrease. In general, given previous levels are frozen (e.g., level 1 & 2), clients that skip a round at level 3 can still train their models at level 4, thus not causing a significant impact.
>
> ### Q2. How does FeMAM with LoRA implement additive composition across levels?
>
> As stated above, the outputs of multiple LoRA adapters at different levels are summed. For a layer with weight $W$, LoRA models the output as:
>
> $$
> h = x \left( W + \sum_{i=1}^{K} B_i A_i \right)
> = xW + \sum_{i=1}^{K} x B_i A_i
> $$

---

> > ### Author Rebuttal · Reviewer_e3kr · 2026-04-02
> >
> > Thank you for the detailed rebuttal. I appreciate the clarifications regarding the additive composition in both classification and LoRA settings, as well as the additional partial-participation results. These responses help improve the paper’s clarity.
> >
> > My main concerns remain. In particular, I still view the contribution as closer to a framework-level composition of existing FL building blocks than a fundamentally new optimization principle, and I remain concerned about the reliance on strong assumptions in the theoretical analysis and the practical inference cost when multiple levels are retained. Overall, while the rebuttal is helpful, I do not think it fully resolves the main issues I raised, so I will keep my score unchanged.

---

### Official Review · Reviewer_sPFp · 2026-03-11

**Soundness:** 4
**Presentation:** 4
**Significance:** 3
**Originality:** 3
**Overall Recommendation:** 5
**Confidence:** 4

**Summary:**

This paper proposes a novel Personalized Federated Learning method by modelling the complex non-IID pattern into a multi-level coarse-to-fine relation. A Federated multi-level additive modelling has been proposed to learn shareable models across levels, and then the personalised model is an additive composition of multiple models across levels.

**Compliance With Llm Reviewing Policy:**

Affirmed.

**Key Questions For Authors:**

Key Questions for the Authors
Please refer to the Weaknesses.
Below are other questions:
1) Can you link the experimental analysis to the methodology part to facilitate reading?
2) Why do you need to disable the thinking mode when using Qwen3-4B?
3) In Table 2, there should be 5*5*5=125 possible combinations. Why does the personalized model only include 50?

**Limitations:**

yes

**Strengths And Weaknesses:**

Strengths
1) The proposed method is novel. A hierarchical structure has been proposed to capture the multi-granularity knowledge across non-IID datasets, and then an additive mechanism has been applied to combine different models into a personalized ensembling model. The whole framework can be explained as a multi-level residual block in Figure 2. The explanation is reasonable and new to the FL domain.
2) The multi-level non-IID is a new practical setting in FL. Most of the previous works only explore the simple patterns among non-IID datasets. Complex pattern across non-IID datasets is a new FL setting. Moreover, if a method can tackle complex patterns, it can also tackle simple patterns, thus achieving a unified solution to FL’s non-IID issues in a broad range of application scenarios.
3) The paper is well written and easy to understand. Each figure is relatively simple and straightforward; thus, they are very easy to understand.  The contents are organized in a logical way. The manuscript of the main body (8 pages) is self-contained. The 9-page appendix provided sufficient details to analyse the method in various ways.
4) The paper’s claims are well-supported by both theoretical analysis and experimental analysis. Many experimental settings are original and comprehensive, which has the potential to be a new benchmark for FL with complex non-IID patterns.
5) The source code has been provided in the supplementary materials that enhanced the reproducibility of this work.

Weaknesses
1) I understand the current learning scenario is based on an abstraction (Figure 1) of the complex learning environment with FL settings. It could be further enhanced by aligning it to more specific examples.
2) In a continual learning scenario, the fixed hierarchical structure might hinder the update of the personalised models. It would be better to discuss how the proposed framework can be adapted to a continual learning scenario in future work.
3) It is reasonable to choose clustered FL as the implementation method for intermediate levels. However, can you discuss whether you can replace the clustering structure with a mixture-of-expert gating mechanism?

---

> ### Author Rebuttal · Authors · 2026-03-28
>
> ### W1. A more concrete real-world example is needed to better ground the abstract FL scenario in Figure 1.
>
> Thanks for the helpful suggestion. As a more specific example, consider the recent rapid growing embodied AI systems in factories and warehouses, where multiple sites collaboratively improve models for robot perception, manipulation, and navigation. At the coarsest level, robots across the whole deployment share common knowledge about safety constraints, object geometry, and generic motion skills. At a finer level, robots with similar roles, such as mobile transport robots, inspection robots, or assembly assistants, benefit from more specialized shared models. At an even finer level, robots working on similar lines or workcells require additional shared refinement because they operate under similar layouts, tools, and task routines. Finally, each individual robot still needs local adaptation due to viewpoint shifts, calibration drift and floor conditions.
>
> ### W2. The applicability of FeMAM to continual learning and possible future extensions should be discussed.
>
> Thanks for the suggestion. Continual learning scenarios pose a potential challenge to FeMAM’s fixed hierarchical structure. As an initial idea, one possible solution is to share low-level models across tasks while retraining high-level models, i.e., enabling the hierarchical structure to grow  “horizontally”.
>
> ### W3. The possibility of replacing intermediate clustered FL levels with an MoE-style gating mechanism should be discussed.
>
> The current K-means clustering provides a hard partition of the model prediction space, while replacing it with MoE would lead to a soft and continuous model prediction space.In FeMAM, this could be beneficial for discovering a more fine-grained hierarchical structure.
> A potential issue is that MoE typically requires a combination of multiple models,whereas the current FeMAM design restricts each client to select only one model per level. As a result, replacing a clustering level with an MoE level may introduce additional level-wise computational cost.
>
> ### Q1. A clearer link between the methodology and the experimental analysis is needed.
>
> Thanks for the suggestion. We will clarify the connection between methodology and experiments. For example, Section 5.1.2 analyzes the impact of combining multiple levels (i.e., FedAvg, K-means, and local models) on performance. Section 5.1.3 evaluates the convergence behavior brought by the boosting-like optimization. Section 5.1.4 studies the effect of structure pruning on enabling an adaptive model structure.
>
> ### Q2. The reason for disabling the thinking mode in Qwen3-4B should be clarified.
>
> We use Flan-T5 tasks for experiments, which are direct natural language instruction-following tasks that do not require complex reasoning. Qwen3-4B enables the thinking mode by default, so it needs to be explicitly disabled.
>
> ### Q3. It should be clarified why the personalized level has only 50 models instead of all possible 5×5×5 combinations.
>
> The personalized level is to assign one model per-client. Since there are 50 clients, the number of personalized models is also 50, the same as the number of clients.

---

> > ### Author Rebuttal · Reviewer_sPFp · 2026-04-04
> >
> > Thank you for your rebuttal. You have addressed all my concerns. I will keep my current score.

---

### Decision · Program_Chairs · 2026-04-30

**Decision:**

Accept (regular)

**Comment:**

This paper got two Accept and one Weak Reject. This submission is competitive, with a clear motivation of an important problem in FL, sound methodology, and comprehensive experiments. The remaining concerns from the negative review include depth of technical contribution in theory and computation cost. Overall, the AC believes that the demonstrated strengths outweigh the limitations.